



# Impact of runoff temporal distribution on ice dynamics.

Basile de Fleurian[1], Richard Davy[2], and Petra M. Langebroek[3]

[1]Department of Earth Science, University of Bergen, Bjerknes Centre for Climate Research, Bergen, NORWAY
[2]Nansen Environmental and Remote Sensing Centre, Bjerknes Centre for Climate Research, Bergen, NORWAY
[3]NORCE Norwegian Research Centre AS, Bjerknes Centre for Climate Research, Bergen, NORWAY

**Correspondence:** Basile de Fleurian (basile.defleurian@uib.no)

**Abstract.** Records of meltwater production at the surface of the Greenland ice sheet have been recorded with a surprisingly high recurrence over the last decades. Those longer and/or more intense melt seasons have a direct impact on the surface mass balance of the ice sheet and on its contribution to sea level rise. Moreover, the surface melt also affects the ice dynamics through the meltwater lubrication feedback. It is still not clear how the meltwater lubrication feedback impacts the long term

ice velocities on the Greenland ice sheet. Here we take a modelling approach with simplified ice sheet geometry and climate forcings to investigate in more detail the impacts of the changing characteristics of the melt season on ice dynamics. We model the ice dynamics through the coupling of the Double Continuum (DoCo) subglacial hydrology model with a shallow shelf approximation for the ice dynamics in the Ice-sheet and Sea-level System Model (ISSM). The climate forcing is generated from the ERA5 dataset to allow the length and intensity of the melt season to be varied in a comparable range of values. Our

simulations present different behaviours between the lower and higher part of the glacier but overall, a longer melt season will yield a faster glacier for a given runoff value. Furthermore, an increase in the intensity of the melt season, even under increasing runoff, tends to reduce glacier velocities. Those results emphasise the complexity of the meltwater lubrication feedback and urge us to use subglacial drainage models with efficient drainage components to give an accurate assessment of its impact on the overall dynamics of the Greenland Ice sheet.

## 1  Introduction

Since the 2000's a large number of studies have pointed towards large increases in the amount of melt recorded at the surface of the Greenland ice sheet (e.g. Steffen et al., 2004; Mote, 2007; Hanna et al., 2008) This is reflected by the fact that during the first two decades of the twenty-first century we have seen the melt record over Greenland beaten four times already (Nghiem et al., 2005; Mote, 2007; Mernild et al., 2009; Tedesco et al., 2011, 2013a; Tedesco and Fettweis, 2020). Ahlstrøm et al. (2017)

identified in southwest Greenland that a shift in the runoff regime took place in 2003 with an 80% increase in runoff for the following decade compare to the period 1976-2002. The changes in the melt season are clearly observed in the distribution of the melt (Zwally et al., 2011; Sasgen et al., 2012) culminating in the 2012 season when the whole surface of the Greenland ice sheet experienced melt at some point during the year (Nghiem et al., 2012; Tedesco et al., 2013a). But even if it is less visible, the length of the melt season has been increasing since the late 70's (Colosio et al., 2021) and that lengthening has a large



impact on the overall melt of the ice sheet, this has been clearly pointed out during the exceptionally long 2010 melt season
which lead to a large amount of total melt (Tedesco et al., 2011).

These changes in the intensity and length of the melt season encourage us to investigate the effect of these changes on the
overall mass balance and flow of the ice sheet. To study the impact of surface melt on ice dynamics, we have coupled the
Double Continuum (DoCo) subglacial hydrology model to the ice flow model in the Ice-sheet and Sea-level System Model

(ISSM Larour et al., 2012). Studies on alpine glaciers, as well as larger ice sheets, have shown that the meltwater produced at
the surface of the ice is routed supraglacialy and englacialy to reach the base of those glaciers (e.g. Seaberg et al., 1988; Catania
et al., 2008; Smith et al., 2015). Once in the subglacial drainage system, these amounts of water have the potential to alter the
water pressure at the base of the ice which is a major driver of glacier sliding (Iken, 1981; Harper et al., 2007; Sole et al.,
2011; Vincent and Moreau, 2016, e.g.). A large water pressure will trigger a high sliding velocity, while a smaller pressure

will lead to a relatively slower glacier (e.g. Bindschadler, 1983). But the complexity of the subglacial drainage system, and the
fact that it can evolve depending on the volume of water that is available means that increasing volume of meltwater would not
necessarily lead to a constant increase in water pressure and glacier speed (Sole et al., 2013; Tedstone et al., 2015). Increasing
water recharge into the subglacial drainage system will lead to an increase in the subglacial pressure until the drainage system
reaches a tipping point and reorganises itself in a more efficient configuration (Walder and Fowler, 1994; Gordon et al., 1998;

Mair et al., 2002). At this point, the more efficient drainage system will allow to drain the provided water at a much lower
pressure and as such will trigger a deceleration of the overlying glacier (e.g. Anderson et al., 2004; Schoof, 2010).

This threshold behaviour leads to seemingly opposing results of an increase in meltwater availability that can be observed
in western Greenland: (*i*) At high elevations, the larger amount of water routed to the bed of the glacier will increase the
subglacial water pressure and lead to faster glaciers (e.g. Zwally et al., 2002; Doyle et al., 2014). (*ii*) At lower elevations, the

increased lubrication could also lead to the prevalence of a more efficient drainage system, leading to lower water pressure in
the subglacial environment and a slower ice flow (e.g. Sundal et al., 2011; Sole et al., 2013; Tedstone et al., 2015).

A few observations, and modeling studies have confirmed that these different behaviours are linked to the elevation of the
measurement and the main subglacialdrainage mode (efficient or inefficient) at this location (Bartholomew et al., 2010; van de
Wal et al., 2015; de Fleurian et al., 2016). The complexity of the meltwater lubrication feedback hampers its inclusion in

Greenland's mass balance projections. The few model simulations that have taken this feedback into account (Shannon et al.,
2013; Fürst et al., 2015) implemented a direct link between meltwater availability and ice velocity. Those simulations however
do not take into account the spatial discrepancies that are observed and the use of those simple run-off/velocity relationships
are questionable (Truffer et al., 2005). Gagliardini and Werder (2018) used a coupled subglacial hydrology model and ice
dynamic model to investigate the response of a glacier in a 40 year simulation. Their simulations focused on the effect of

an increase in the intensity of the melt season while keeping its duration constant. It is however unclear if both those factors
– melt season length and amplitude – have the same impact on the subglacial drainage system and glacier velocities. The
present study focuses on shorter time scales to provide an answer to this question and a better understanding of the meltwater
lubrication feedback mechanisms. An improved treatment of this feedback is indeed crucial has the recent study of Gagliardini





and Werder (2018) showed that it could account for a volume loss significantly higher than what was estimated from the simple
parameterisation of Shannon et al. (2013); Fürst et al. (2015).

   We will first give an overview of the component of the model which are specific to this study before describing in more
details the set-up and forcing that are used in the study. We first focus on the results from the reference simulation which give
us the opportunity to assess the behaviour of the system before the impact of the different forcings are presented. Finally we
give a broader interpretation of the results of our experiments in light of recent findings on the meltwater lubrication feedback.

## 2   Methods

### 2.1   Model

In order to investigate the impact of meltwater availability on ice dynamics we perform coupled subglacial hydrology and
ice dynamics simulations within the Ice-sheet and Sea-level System Model (ISSM Larour et al., 2012). Within the ISSM,
the ice flow is treated following the Shallow Shelf Approximation (SSA) (Morland, 1987; MacAyeal, 1989). The choice of
this approximation is motivated by a need to run on relatively short time-steps, and so it is necessary to have a relatively
computationally-cheap ice flow model. Since our interest is in the sliding of the glacier (rather than its deformation) SSA is
well suited. The lubrication feedback and the impact of subglacial water on ice flow dynamics depends on the choice of the
sliding law linking the ice dynamics to the subglacial hydrology. Here we use a non-linear friction law as described by Schoof
(2005) and Gagliardini et al. (2007) which links sliding velocities ($\boldsymbol{u_b}$) and basal shear stress ($\boldsymbol{\tau_b}$):

$$\boldsymbol{\tau_b} + \frac{CN|\boldsymbol{u_b}|^{(1/n-1)}}{(|\boldsymbol{u_b}| + C^n N^n A_s)^{(1/n)}}\boldsymbol{u_b} = 0. \tag{1}$$

   Where the parameters $A_s$ and $C$ are the sliding parameter in the absence of cavities and Iken's bound parameter respectively.
Iken's bound (Iken, 1981) represent the maximum value that can be taken by $\tau_b/N$ and is only determined by the maximum
up-slope of the bed. $N$ is the effective pressure which is produced by the subglacial hydrology model in response to the surface
melt forcing. The subglacial hydrology model used is the Double Continuum (DoCo) approach as described in de Fleurian
et al. (2016). This model is based on a double sediment layer system. The first layer, with a low conductivity ($K_s$) represents
the inefficient drainage system (IDS). The second layer, with a much higher conductivity ($K_e$) is only activated when the local
effective pressure is equal to 0. Once activated, the thickness ($e_e$) of this efficient drainage system (EDS) evolves from its initial
thickness following equations describing the size of subglacial channels (Röthlisberger, 1972; Nye, 1976) and scaled to take
into account the specific geometry of the EDS.

$$\frac{\partial e_e}{\partial t} = \frac{g\rho_w e_e K_e}{\rho_i L}(\nabla h_e)^2 - 2An^{-n}N^n e_e, \tag{2}$$

   This equations involves the widening of the system as the ice melts where $\rho_w$ and $\rho_i$ are the densities of freshwater and ice,
$K_e$ and $e_e$ the conductivity and thickness of the EDS $h_e$ the water head in the EDS and finally $L$ the latent heat of fusion of





**Table 1.** Values of the model parameters.

| Symbol | Parameter | Value |
|--------|-----------|-------|
| $e_s$ | IDS thickness | $20$ m |
| $e_e$ | EDS initial thickness | $5.0 \times 10^{-3}$ m |
| $K_s$ | IDS conductivity | $2.0 \times 10^{-3}$ ms$^{-1}$ |
| $K_e$ | EDS conductivity | $9.0 \times 10^{1}$ ms$^{-1}$ |
| $\omega$ | porosity | $0.4$ |
| $\gamma$ | leakage time | $1.0 \times 10^{-9}$ s$^{-1}$ |
| $A_s$ | Sliding Parameter | $3.2 \times 10^{-21}$ mPa$^{-3}$s$^{-1}$ |
| $C$ | Iken's Bound | $0.35$ |
| $\rho_w$ | water density | $1{,}000$ kgm$^{-3}$ |
| $\rho_i$ | ice density | $910$ kgm$^{-3}$ |
| $g$ | gravitational acceleration | $9.8$ ms$^{-2}$ |
| $L$ | latent heat of fusion for the ice | $3.34 \times 10^{5}$ Jkg$^{-1}$ |
| $A$ | Glen's flow law parameter | $6.34 \times 10^{-25}$ Pa$^{-1}$s$^{-1}$ |
| $n$ | Glen's flow law exponent | $3$ |
| $\mu$ | water viscosity | $1.78 \times 10^{-3}$ Nsm$^{-2}$ |
| $\beta_w$ | water compressibility | $5.0 \times 10^{-10}$ Pa$^{-1}$ |

ice. The other term represents the closing of the system through ice creep where $A$ and $n$ are Glen's parameter and exponent. As the pressure in the EDS decreases, it will get thinner until the point where its transmitivity ($T_e = K_e \times e_e$) is lower than that of the IDS at which point the EDS is deactivated.

The parameters of the model are given in Table 1, and detailed information about the model can be found in de Fleurian et al. (2016) and de Fleurian et al. (2014).

One challenge when running a coupled ice-flow subglacial hydrology model is that the two systems are responding on different timescales. The subglacial hydrology model needs short time-steps to achieve stability and provide reliable results while the ice flow model can run on a longer time-step, which saves on computing time. In this study we used a 15 minutes time-step for the hydrology model and a one hour stepping for the ice flow model. Testing of a number of different options showed that this specific combination of time-steps gave consistent results while keeping the computational cost at a manageable level. The management of the different time-steps for the coupling is performed through the averaging of the effective pressure over the length of the ice flow time-step and the averaged effective pressure is then used to compute the flow. The geometry of the ice model needed as an input to the hydrology model is then kept fixed for the four sub-steps of the subglacial hydrology model.



## 2.2 Geometry and spin-up

The geometry of the system presented on Fig. 1 is kept as simple as possible in order to obtain a consistent response to applied perturbations. The domain on which the simulations are performed is a synthetic representation of a land terminating ice sheet

margin. The glacier is 150 km long and 20 km wide with a flat bedrock of elevation $z_b = 465$ m and an initial surface elevation $z_s$ defined by a parabolic function:

$$z_s(x,y) = 4.5 \times \sqrt{x + 4000} + 186 \tag{3}$$

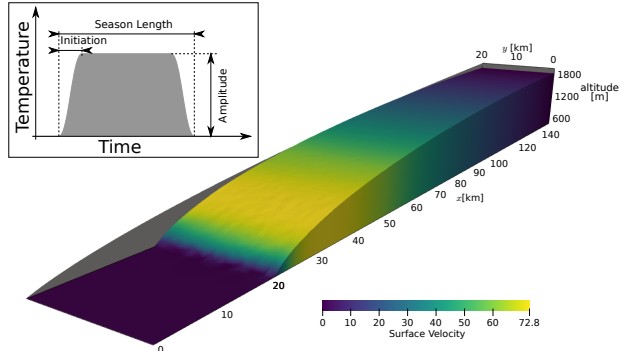

**Figure 1.** Initial parabolic geometry in grey in the background and final geometry after spin-up. The colour represent the mean annual velocities at the end of the spin-up simulation. The inset shows the shape of the temperature forcing used and the different parameters which impact are studied.

The parabolic function is chosen such that it resembles the topography of West Greenland as used in the ERA5 dataset. From this geometry, a long spin-up is run to achieve stability of the different components of the system. This involved an offline

coupling of the ice sheet model where the surface mass balance was given by the reference forcing described in Sect. 2.3 and the subglacial hydrology model. A relatively stable state with only a small volume loss was attained after roughly 5000 years of simulation. This final state is then used for all the perturbation simulations.

## 2.3 Forcing

Here we take an idealised forcing using the method first described by Hewitt (2013). In this formulation the temperature is first

defined at a reference height using the equation:

$$T_{ref}(t) = \frac{r_m}{\Delta_m} \times \left( \frac{1}{2} \tanh\left( \frac{t - t_{spr}}{\Delta_t} \right) - \frac{1}{2} \tanh\left( \frac{t - (t_{spr} + \Delta_m)}{\Delta_t} \right) \right) \tag{4}$$

where

- $T_{ref}$ is the temperature at the reference elevation of 465 m at the front of the glacier [° C].





- – $\Delta_m$ is the length of the melt season [days].

- – $r_m$ is the positive degree day at the reference elevation [° Cdays].

- – $t_{spr}$ is the beginning of the melt season (day 100)

- – $\Delta_t$ is the length of the initiation of the melt season [days].

The runoff itself is then computed using a given lapse rate

$$r(s,t) = max\left\{0, T_{ref}(t) \times (z_s - 465) \times r_s\right\} \times ddf \tag{5}$$

where $r_s$ is the lapse rate in $° \, \mathrm{Km}^{-1}$, $ddf$ is the degree day factor or conversion rate from temperature to runoff.

The three parameters of this model that we chose to test the sensitivity are the length of the melt season ($\Delta_m$), the PDD at the reference elevation ($r_m$), and the length of the initiation ($\Delta_t$) as shown on the inset of Fig. 1. We use the ERA5 reanalysis data to derive a realistic positive degree day (PDD), season length and lapse rate for our model (Hersbach et al., 2020). First we extracted the daily surface air temperature from 1979 to 2018 for south-western Greenland at a fixed latitude and for a
longitude band going from close to the coast (465 m above sea level) to near the highest point of the land at 2256 m above sea level (67° N, 45–50° W). To calculate the length of melt season for each year we smoothed the daily temperatures using a 15-day midpoint running mean. This smoothing was applied to avoid creating anomalously long melt seasons due to a single day of high temperatures occurring far outside of the rest of the melt season of a given year. The length of the melt season was then calculated using the first and the last day of the year with daily-mean temperatures greater than 0° C. The lapse rate for
each day was calculated by taking the gradient of a least-squares linear regression of temperature against elevation across the domain. The daily lapse rate was then averaged for the every day where the temperature was above 0° C at at least one grid point, to obtain a typical lapse rate for the melt season. To calculate the positive degree days (PDD) for each year we simply summed the positive temperatures in Celsius over all *n* levels of the ERA5 reanalysis data within our domain:

$$PDD = \sum_{s=1}^{n} max\left(0, T(s)\right) \tag{6}$$

This procedure gave us a single representative value for the PDD, length of the melt season, and the typical lapse rate for each year from 1979-2018. We then fit a normal distribution to the values for all years to determine the mean and standard deviation (SD) of the inter-annual variability in each of these variables. These were used to define low (mean-SD), medium (mean), and high (mean+SD) values for our sensitivity studies. We also tested the steepness of the temperature variation at the beginning and end of the melt season, which is controlled by the parameter $\Delta_t$ and called initiation further on in the manuscript (Fig. 1).
Since this cannot be derived from ERA5 data we instead chose these values pragmatically to create a spread of reasonable seasonal cycles. The different initiation and melt season length are given in Table 2. The value for $r_m$ is presented there with the ratio $r_m/\Delta_m$ which represent the maximum temperature at the reference elevation.

Regardless of the length of the melt season we define the summer as the period between days 100 ($t_{spr}$, April $10^{th}$) and 241 ($t_{spr} + \Delta_m$) for the reference simulation, August $29^{th}$ of the simulation.



**Table 2.** Values of the different parameters used in the perturbation experiments

| Parameter | low value | reference value | high value |
|---|---|---|---|
| $\Delta_m$ [days] | 109 | 141 | 173 |
| $r_m/\Delta_m$ [° C] | 5.40 | 5.85 | 6.47 |
| $\Delta_t$ [days] | 5 | 10 | 15 |

For the simulations with reference PDD, $r_m/\Delta_m$ values for the long and short initiation are respectively 5.96 and 5.74° C

## 2.4 Statistics

Due to the non linearity of the friction law, combined with the threshold present in the activation of the EDS system, the model results have a significant spread for similar simulations. To extract the characteristic evolution of the system we perform an ensemble of simulations for each parameter set. Each ensemble contains 100 simulations which only differ in their starting time which is shifted 1 minute for each simulation. This sampling allows us to get similar simulations while still keeping the natural variability of the model. The different starting time leads to every simulation having results on slightly different time-steps. To remedy this problem, and to ensure that the results are not shifted in time, all the simulations are synchronised on the same time-steps through linear interpolation before any further analysis. The distribution of the simulations that are produced through this procedure do not yield a normal distribution as seen on Fig. 2. To overcome this distribution issue, we decided to use the Wilcoxon signed-rank test to investigate the difference of every parameter set to the reference simulation. We use a non parametric Wilcoxon signed-rank test to assess the differences between the reference simulation and the perturbed simulation and test if the perturbation lead to significant differences in the response of the model. Every time that the differences from one simulation with respect to the reference is said to be significant it means that the null hypothesis of the Wilcoxon signed-rank test (the median of the differences is zero) is rejected with a 1% confidence level.

## 3 Results

### 3.1 Reference Simulation

Figure 2 shows the evolution of the mean runoff, ice velocity and effective pressure together with the point of highest active EDS throughout the simulations. The highest active-EDS point corresponds to the surface elevation of the farthest inland location where the EDS is active at any time in the simulation. This variable is a good indicator of both the spread of the EDS but also of the dynamics of its development and collapse throughout the seasons.

On this figure we show each individual simulation as a grey line and the mean value of the ensemble is represented by the thick black line. The general evolution of the mean velocities on the whole domain is similar for every year of the simulation as shown on Fig. 2b. The first year of the simulation shows a slightly lower spring speed-up event which is related to the



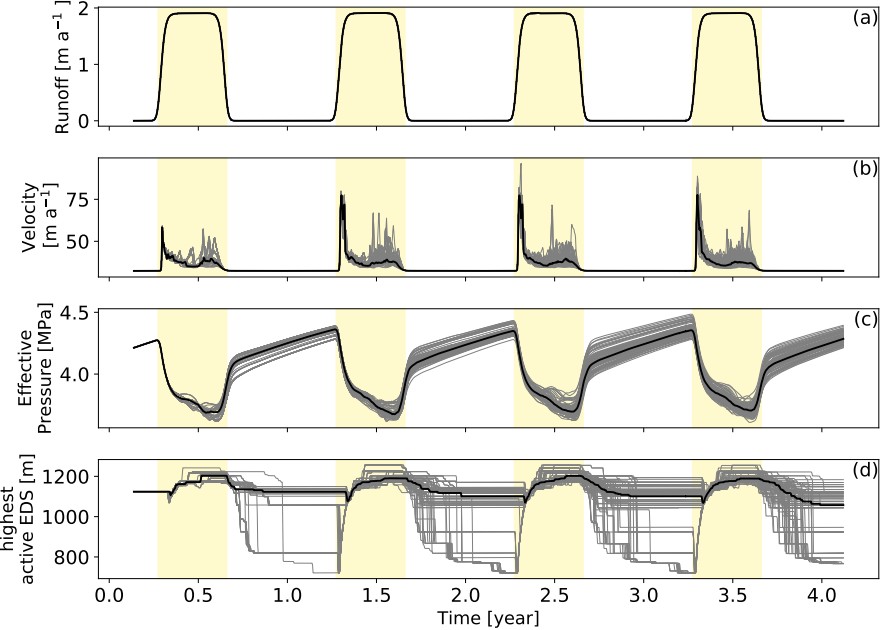

**Figure 2.** Evolution of the surface runoff (a), ice velocity (b), effective pressure (c) and highest active efficient drainage system node (d) averaged over the whole domain. The yellow shading represent the summer period while every grey line corresponds to one member of the ensemble of simulations and the black thick line represent the median of the ensemble.

relaxation of the geometry from the spin-up. In the following we will focus on the second year of the simulation which is the one that was perturbed for the sensitivity experiments. As the melt season starts (yellow shaded regions of Fig. 2) runoff

increases, building up basal water pressure under the ice and triggering a drop in effective pressure leading to a sharp speed up of the glacier. The decrease in the effective pressure (Fig. 2c) slows down as the efficient drainage system develops and it drains higher regions of the glacier (Fig. 2d). The velocities then quickly subside as the efficient drainage system develops in the lower part of the glacier and extends upstream. After this first speed-up event, the velocities tend to stabilise to a lower level with each simulation showing some later velocity spikes which tend to occur more often at the end of the melt season.

These higher velocities are due to the continuous lowering of the effective pressure throughout the melt season. These spikes are linked to a secondary reduction in the effective pressure which happens when the EDS is fully developed and so the water going into the system tends to overload this system. These accelerations have been observed in Greenland where it is usually linked to more important melt or rainfall events after the development of an efficient drainage system (e.g. Cowton et al., 2013; Doyle et al., 2015; van de Wal et al., 2015). We will further refer to this late acceleration phase as the autumn acceleration. At

the end of the melt season, the drop in runoff leads to a fast increase in the effective pressure which goes back to its winter level. At the lower elevations (Fig. 3g-h), we see an overshoot of those winter values as the subglacial drainage capacity is higher than the recharge at this time.

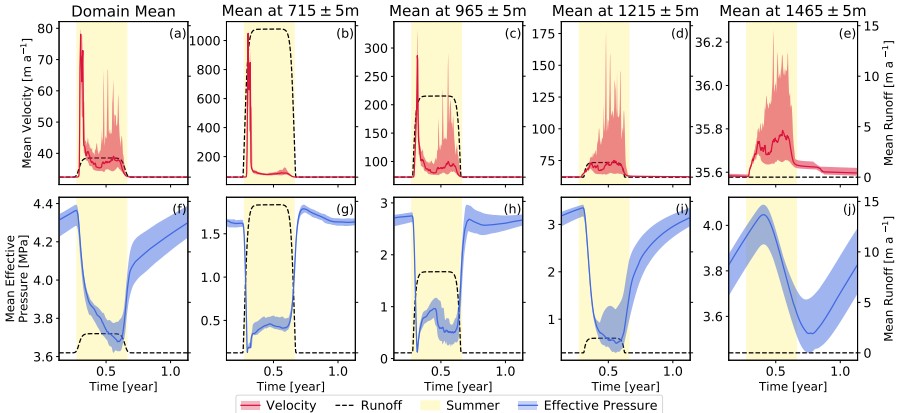

**Figure 3.** Evolution of the velocity (a-e) and effective pressure (f-j) for the second year of the simulation. The different columns present values at different altitudes and the mean value on the domain is given for reference. The black dashed line shows the runoff at each elevation on the right axis, the range of the left axis (for velocity and effective pressure) varies in the different panels. The yellow shading represents the summer as defined in Sect. 2.3

The evolution of the mean variables over the whole glacier are however not representative of their local evolution. In Fig. 3 we show the evolution of the velocity and effective pressure at four different altitudes on the domain. These figures show two main behaviours for those two variables. The lowest part of the glacier, closer to the front (at 465 m of altitude), is characterised by a strong spring speed-up (Fig. 3b-c), whereas the higher region of the domain shows a more gradual increase of velocities throughout the melt season (Fig. 3d-e). These different responses in velocity are driven by the effective pressure at the base of the glacier. At the lower elevations (Fig. 3g-h), the effective pressure shows a sudden drop at the beginning of the melt season followed by a quick rebound when the efficient drainage system activates, which drives the spring speed-up event. At higher elevations (Fig. 3i) the decrease in effective pressure is more gradual and it does not reach values low enough to trigger a spring speed-up event or lead to the activation of the efficient drainage system. Even higher up on the glacier the effective pressure is driven by downstream activity as there is no runoff at these elevations. As a result, the effective pressure shows very small variations which get more and more out of phase with the melt season as we go higher up the glacier (Fig. 3j). We will in the following use the acceleration of the glacier as a reference metric to define the elevation at which the glacier shifts from the spring speed-up driven velocity pattern to the more gradual one. This shift in behaviour (henceforth called $SSU_{max}$) is set from the reference simulation ensemble as the altitude at which the glacier acceleration at the beginning of the melt season drops under $20 \, \mathrm{ma}^{-1}\mathrm{d}^{-1}$ which is shown by the white line on all panels of Fig. 4. In the reference simulation, $SSU_{max}$ is encountered at 1058 m and throughout the different simulations its elevation varies between 1000 and 1150 m.

We can also observe the two different behaviours of the system in Fig. 4. Here we see that the fast spring speed-up is confined bellow the $SSU_{max}$ elevation (white line on all panels) as expected from its definition (Fig. 4b). The EDS transmitivity (Fig. 4d) is a proxy for the capacity of the subglacial drainage system to drain water. The white region corresponds to periods where the EDS is not active, while the higher values indicate a highly developed and efficient system. The lowering of the transmitivity at



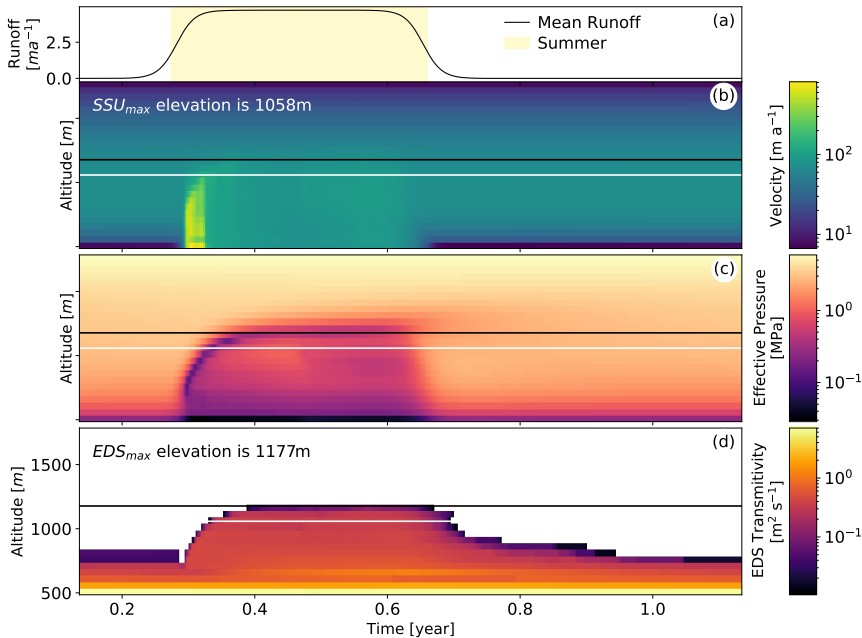

**Figure 4.** Evolution of the velocity (b), effective pressure (c) and efficient drainage system transmitivity (d) binned in 50m elevation bands. The runoff is given in panel (a) for reference. The white line across panels (b) to (d) represents $SSU_{max}$ as defined in Sect. 3.1

the end of the melt season shows how a drop in water pressure leads to the contraction of the EDS and ultimately to its collapse and deactivation. From the EDS transmitivity we can define the maximum altitude at which the EDS is activated ($EDS_{max}$).

$EDS_{max}$ is slightly higher than $SSU_{max}$ and is the effective boundary between the lower part of the glacier where the effective pressure is mostly controlled by the efficient drainage system, and the upper part of the glacier where the effective pressure evolution shows more gradual evolution in line with the weaker conductivity of the inefficient drainage system.

### 3.2 Melt Season Length Forcing

We first investigate the effect that a shorter or longer melt season has on the glacier's velocities. Since we chose to keep the

runoff constant for this set of simulations, changes in the length of the melt season simultaneously impact the melt intensity. This choice allows to compare both length and intensity of the melt season and investigate which parameter has the stronger effect on the glacier's velocity. In Sec. 3.3 we will present more details about the specific impact of a change in melt intensity or melt season length when they are applied separately. Figure 5 presents the comparison between the reference simulation (grey), the long and low intensity melt season (red) and the short and high intensity melt season (blue) is shown for two

different altitudes.

Starting with the velocities we see quite different evolution for the long and short melt seasons with a distinct evolution above and bellow the $SSU_{max}$ elevation. We must note here that $SSU_{max}$ also varies with the different forcing ranging from





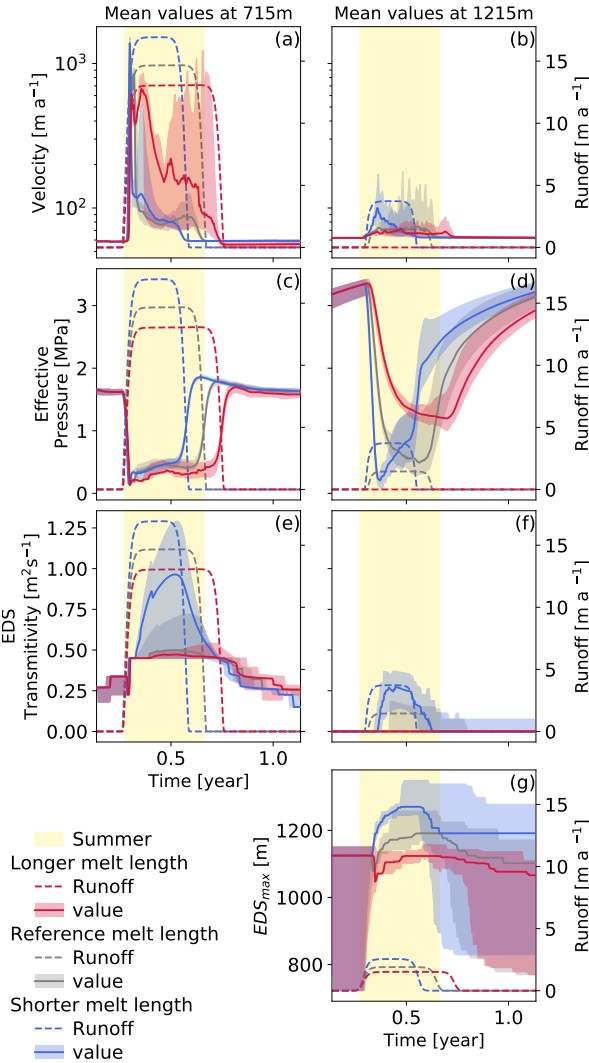

**Figure 5.** Evolution of the velocity (a-b), effective pressure (c-d) and efficient drainage system transmitivity (e-f) at two different altitudes. Evolution of the maximum height of the active EDS (g). Every panel shows the reference simulation in grey, the longer lower intensity melt season in red and the shorter higher intensity melt season in blue. The same colour scheme is used for the runoffs which are shown as dashed lines and are referenced on the right axis. The yellow shading correspond to the fixed duration summer as defined in Sect. 2.3

996 m for the longest melt season to 1100 m for the shortest one as reported in Table 3. At low elevation, the shorter and more intense melt season leads to a sharper, shorter lived, but also more intense spring speed-up (Fig. 5a). This differs with the evolution modelled for the longer melt season where the spring speed-up is less marked initially and followed by a second acceleration event. Contrasting with that, the end of the melt season and the begining of winter show an acceleration for some of the simulations forced by the long melt season as seen by the red shading on Fig. 5a whereas the early shutdown of the short





melt season leads to slow velocities at the end of summer with no simulations showing re-acceleration at this time. At 1215 m, which is above $SSU_{max}$ for all simulations, the velocity evolution is quite different. While the reference simulation was only

showing a small increase in velocities, there is a quite large acceleration for the intense melt season with some simulations showing high velocities throughout the melt period. This contrasts with the reference simulation where there was no speed-up at the initiation of the melt season but a more gradual build-up of velocities until the glacier reached its maximum velocities at the end of the melt season. At the other end of the spectrum, the long but less intense melt season leads to smaller velocities than the reference at the beginning of the melt season but this low acceleration is sustained throughout the melt period. The

differences that are observed in the velocity evolution for the different altitudes and melt season length are explained by the evolution of the effective pressure which is tightly linked to the efficiency of the draining system. Below $SSU_{max}$, the effective pressure shows a drop early in the melt season for both the perturbed scenarios. However, the effective pressure recovers faster for the shorter (and more intense) melt season where it quickly rebounds to a summer value plateau. This contrasts with the short melt season where the summer plateau is reached later and not after a second reduction in effective pressure, due to the

second acceleration event. The EDS transmitivity is responsible for these dynamics, whereby the higher intensity of the shorter melt season allows the glacier to quickly develop a very efficient subglacial draining system which enables a fast rebound of the effective pressure. The greater efficiency of the EDS for the short melt season is also apparent in the $EDS_{max}$ evolution (Fig. 5g) where the shorter melt season leads to a faster development of the EPL towards higher elevations than the one reached in simulations with longer melt seasons. We note also that with a more developed efficient drainage system (Fig. 5e-f)

the overshoot which is observed at the end of the melt season is more marked. This overshoot leads to the lowest velocities being reached just at the end of the melt season and the ice flow then accelerates throughout the winter.

At higher elevations, the changes in the length and intensity of the melt season leads to contrasted evolution for the effective pressure. For the short melt season, the response is similar to the one that was observed at lower elevation. Indeed, at this elevation the runoff is quite substantial leading to the rapid decrease of the effective pressure at the beginning of the melt

season. The effective pressure then reaches values that are low enough to activate the EDS as seen in Fig. 5f-g, which in turn drives the strong increase in effective pressure. In terms of velocities, the variations in effective pressure drive a fast flow at the beginning of the melt season which quickly subsides. The reference and longer melt season show quite a different behaviour regarding their effective pressure. For those simulations, the runoff at this altitude is too low to trigger the activation of the EDS. This leads to a gradual decrease of the effective pressure throughout the melt season which only increases again when

the melt season ends. This pressure evolution induces moderate summer velocities that last throughout the melt season.

Table 3 summarises the impact of the length of the melt season on the dynamics of the glacier. The shorter and more intense melt season shows a significant reduction in the overall glacier velocity which is mostly driven by a reduction of the summer velocities below the $SSU_{max}$ altitude (1071 m in this case). In the case of the long melt season, the acceleration of the glacier is mostly driven by the summer acceleration of the lowest parts of the glacier. There is a slight deceleration at higher elevations

but that is not sufficient to offset the doubling of the mean summer velocities that is observed during summer at the lowest elevations. The winter mean velocities of the longer melt season experiments are somewhat deceiving as the longer melt season means that part of the melt is now happening during the so called winter. Coming back to Fig. 5a it is clear that the winter



**Table 3.** Velocity difference to the reference simulation for different melt season length and intensity. The values in bold fonts are the simulations for which the difference with respect to the reference simulation is significant as per a Wilcoxon on test with p=0.01. The summer and winter are defined as fixed periods with summer starting on day 100 and ending on day 241 while winter covers the rest of the year.

|  |  | Annual mean | Summer Mean | Winter Mean |
|---|---|---|---|---|
| Shorter melt season higher intensity $SSU_{max}$ = 1071 m $EDS_{max}$ = 1251 m | domain | **-4.13%** | **-7.77%** | **-0.17%** |
|  | 715 m | **-6.55%** | **-12.13%** | 0.03% |
|  | 965 m | **-3.23%** | **-6.75%** | **-0.32%** |
|  | 1215 m | -0.02% | 0.36% | **-0.34%** |
|  | 1465 m | **0.15%** | **0.26%** | **0.09%** |
| Longer melt season lower intensity $SSU_{max}$ = 1003 m $EDS_{max}$ = 1123 m | domain | **20.98%** | **32.55%** | **9.55%** |
|  | 715 m | **66.13%** | **115.12%** | **5.14%** |
|  | 965 m | **7.54%** | **10.71%** | **5.50%** |
|  | 1215 m | **-1.38%** | **-5.02%** | **1.39%** |
|  | 1465 m | **-0.04%** | **-0.12%** | 0.00% |

velocities in this case are significantly lower than in the other simulations, which is in line with the behaviour shown by Sole et al. (2013) in southwest Greenland. It is interesting to note here the reversed pattern in the velocity variations with the altitude,

where the shorter and more intense melt season leads to a slower glacier at low elevation and faster at higher elevations where the reversed response is observed for a longer, less intense melt season. In both cases however it is the summer velocities at low elevations which are driving the changes in the annual velocities of the glacier.

### 3.3 Intensity vs length

To discriminate between intensity and length of the melt season the requirement that the runoff is to be equal in all simulations

has been released. First we run a series of experiments with the same intensity but in which the length of the melt season varies (Fig. 6). Second, we keep the melt season length fixed, but vary the intensity (Fig. 7).

As expected the evolution at the beginning of the melt season of those experiments is very similar as they are all experiencing the same forcing. The simulations start to differ at the point when the short melt season ends. At this time, the effective pressure for this simulation goes back to its winter level with a slight overshoot (Fig. 6c). This timing also coincides with a period during

which the effective pressure of the two longer melt seasons are dropping again. This decrease seems to be due to the efficiency of the drainage system reaching a maximum before slowly decreasing and driving the effective pressure down (Fig. 6e). The effect of this effective pressure decline is transferred to the ice velocities where we see an accelerating trend at the end of summer for the simulations with the longer melt seasons (Fig. 6a). At higher elevations the overall behaviour of the glacier is not strongly impacted by the change in melt season length. Whereas at lower elevations, when the simulation with the

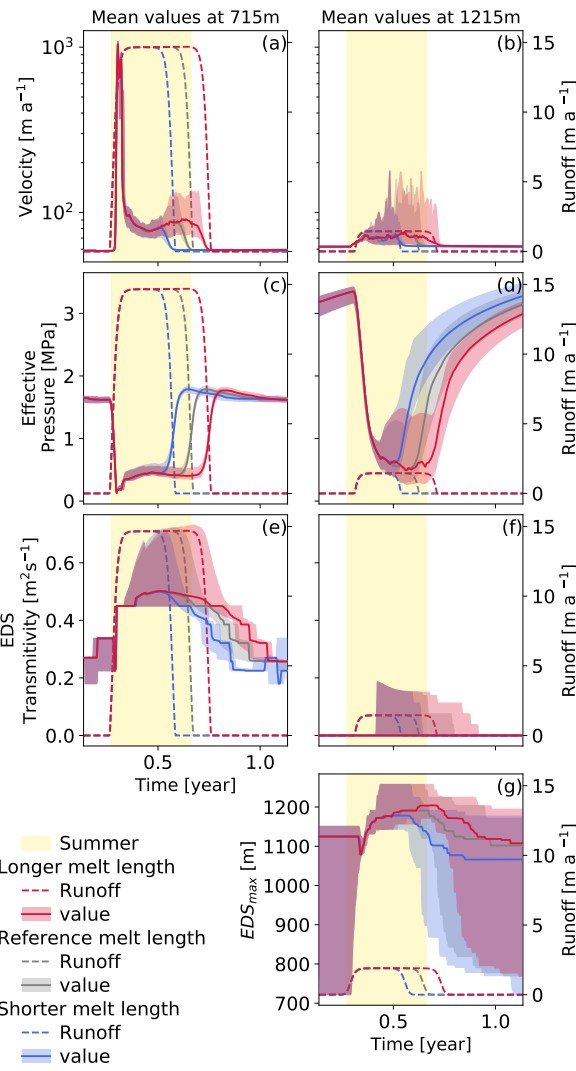

**Figure 6.** Evolution of the velocity (a-b), effective pressure (c-d) and efficient drainage system transmitivity (e-f) at two different altitudes. Evolution of the maximum height of the active EDS (g). Every panel shows the reference simulation in grey, the longer melt season in red and the shorter melt season in blue. The same colour scheme is used for the runoffs which are shown as dashed lines and are referenced on the right axis. The yellow shading correspond to the fixed duration summer as defined in Sect. 2.3

shortest melt season finishes the melt season the effective pressure goes back to its winter levels, while the simulations with longer melt seasons have a continued decrease of the effective pressure (Fig. 6d). In terms of velocity, only the duration of the summer acceleration is significantly different under the different forcings, with a mean summer velocity that is comparable for all simulations (Fig. 6b). However, the extreme values for the longer melt seasons tend to show more important acceleration events happening at the end of the melt season.





**Table 4.** Velocity difference to the reference simulation for different melt season length and reference intensity. The values in bold fonts are the simulations for which the difference with respect to the reference simulation is significant as per a Wilcoxon test with p=0.01. The summer and winter are defined as fixed periods with summer starting on day 100 and ending on day 241 while winter covers the rest of the year.

| | | Annual mean | Summer Mean | Winter Mean |
|---|---|---|---|---|
| Shorter melt season $SSU_{max} = 1058$ m $EDS_{max} = 1164$ m | domain | **-4.47%** | **-8.38%** | **-0.17%** |
| | 715 m | **-2.80%** | **-5.55%** | **0.39%** |
| | 965 m | **-2.79%** | **-5.94%** | **-0.16%** |
| | 1215 m | **-1.90%** | **-4.17%** | **-0.33%** |
| | 1465 m | **-0.07%** | **-0.11%** | **-0.05%** |
| Longer melt season $SSU_{max} = 1058$ m $EDS_{max} = 1203$ m | domain | **4.45%** | **2.45%** | **8.45%** |
| | 715 m | **2.96%** | **2.52%** | **5.03%** |
| | 965 m | **2.87%** | **2.92%** | **3.30%** |
| | 1215 m | **1.33%** | **1.68%** | **1.22%** |
| | 1465 m | **0.05%** | **0.03%** | **0.07%** |

The overall changes in velocities are presented in Table 4, contrasting with the experiments performed with a constant PDD, we see here that the changes in the melt season length act in the same direction (either acceleration or slow down) at all altitudes. We also note that the symmetry in the forcing difference leads to a linear response in term of velocities where the annual mean velocities for the shorter melt season forcing are decreased by a similar amount with respect to the reference simulation and vice versa.

Comparing the simulations with different intensities yields more significant differences between simulations (Fig.7). We find here a similar pattern to the simulations with varying intensity and duration (Sec. 3.2). At low elevation the initiation of the melt season yields the same pattern with a strong spring speed-up which is followed by a secondary acceleration in the case of the low intensity forcing (Fig.7a). The higher intensity melt season also shows a marked overshoot of the winter effective pressure value at the end of the melt season (Fig.7c). Above $SSU_{max}$ the difference in the melt intensity means that the experiments with

the lower intensity do not experience runoff at this elevation. This leads to the effective pressure in this case being driven by the downstream evolution of the effective pressure which results in a slow decrease of the effective pressure throughout the melt season (Fig.7d). The velocity response follows the variations in effective pressure with a slow increase towards a rather low maximum summer velocity towards the middle of the melt season (Fig.7b). Despite showing some recharge at this elevation, the reference simulation shows a similar velocity pattern as the meltwater availability at this altitude does not lead to very low

water pressures. The response to the more intense (warmer) melt season is quite different. Here the water input is sufficient to drive a quick drop in effective pressure and the following rebound once the efficient drainage system is activated (Fig.7f). In



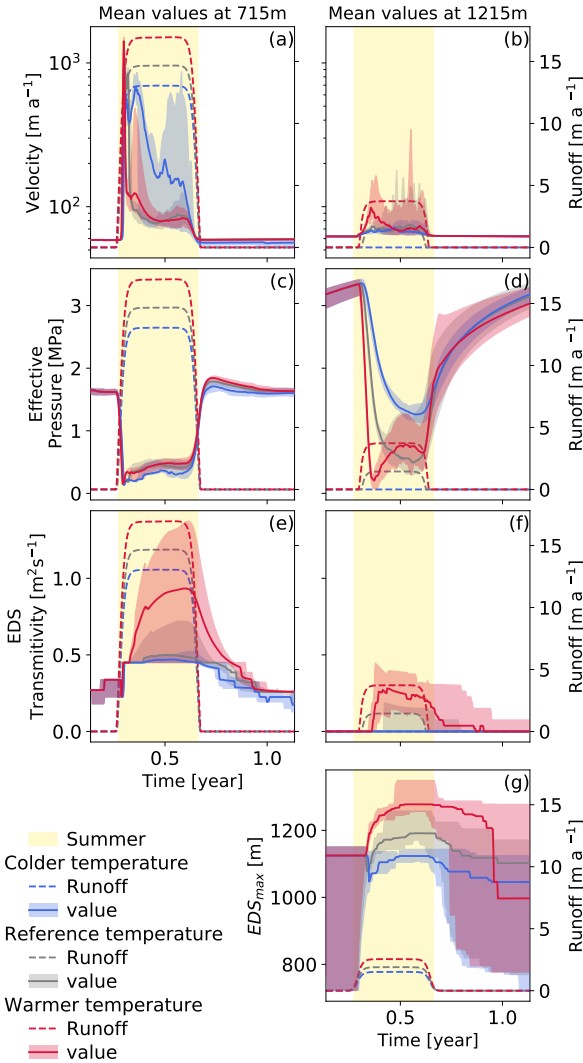

**Figure 7.** Evolution of the velocity (a-b), effective pressure (c-d) and efficient drainage system transmitivity (e-f) at two different altitudes. Evolution of the maximum height of the active EDS (g). Every panel shows the reference simulation in grey, the higher intensity melt season in red and the lower intensity melt season in blue. The same colour scheme is used for the runoffs which are shown as dashed lines and are referenced on the right axis. The yellow shading correspond to the fixed duration summer as defined in Sect. 2.3

term of velocities, that translates to a spring acceleration of small magnitude with gentler slopes than what is observed on the lower part of the glacier.

The general impact of the melt season intensity on the glacier's velocity shows counter-intuitive results (Table 5). The mean velocities over the whole domain indicate here that the stronger intensity forcing does not lead to any significant difference in velocity compared to the reference simulation. However, the colder (lower intensity) melt season shows a substantial increase





**Table 5.** Velocity difference to the reference simulation for different intensity of the melt season with the reference duration. The values in bold fonts are the simulations for which the difference with respect to the reference simulation is significant as per a Wilcoxon test with p=0.01. The summer and winter are defined as fixed periods with summer starting on day 100 and ending on day 241 while winter covers the rest of the year.

|  |  | Annual mean | Summer Mean | Winter Mean |
|---|---|---|---|---|
| Colder temperatures $SSU_{max}$ = 1003 m $EDS_{max}$ = 1111 m | domain | **13.62%** | **26.05%** | **-0.13%** |
|  | 715 m | **45.15%** | **94.95%** | **-4.60%** |
|  | 965 m | **4.16%** | **8.30%** | **0.34%** |
|  | 1215 m | **-2.31%** | **-5.53%** | **-0.04%** |
|  | 1465 m | **-0.08%** | **-0.14%** | **-0.05%** |
| Warmer temperatures $SSU_{max}$ = 1077 m $EDS_{max}$ = 1269 m | domain | -0.44% | -0.85% | -0.00% |
|  | 715 m | -4.93% | -8.64% | **-0.56%** |
|  | 965 m | **-2.71%** | **-5.82%** | **-0.17%** |
|  | 1215 m | **2.25%** | **5.49%** | -0.03% |
|  | 1465 m | **0.35%** | **0.64%** | **0.22%** |

in the mean annual velocities. This acceleration is mostly due to a two fold increase of the summer velocities at low elevation, which in our experiments can not be offset by the slowdown at higher elevation or during winter. The increase of the melt area that is driven by the larger intensity of the melt season drives an acceleration of the upper part of the glacier, but this
acceleration is not sufficient to impact the overall velocity of the glacier.

Figure 8 summarises the effect of intensity and length of the melt season on both the subglacial hydrology system and the glacier dynamics. In Fig. 8a we see a clear linear relation between the increase in the runoff volume and the transmitivity of the efficient drainage system, that shows that on the whole glacier, an increase in meltwater volume leads to a more developed subglacial drainage system either in areal extent (increase of $EDS_{max}$) or transmitivity of the system. This relationship is not
related to the distribution of the melt throughout the year, and we see that an increase in runoff driven by either a longer melt season (growing marker sizes) or a intensifying melt season (darkening markers) lead to a similar increase in transmitivity.

The response is more contrasted for the areal development of the efficient drainage system. This characteristic is described by $EDS_{max}$ (Fig. 8b), the maximum altitude of the glacier surface under which the efficient drainage system is active. As for the transmitivity, Fig. 8b shows that the efficient drainage system develops further upstream if the increase in runoff is
due to an increase in the intensity of the melt season. However, $EDS_{max}$ shows a more gradual increase if the runoff is only increased by a lengthening of the melt season. A result of these two different behaviour is that for a given runoff, the spread of the efficient drainage system will be greater for a short and intense melt season (small black dot of Fig. 8b) than for a long and colder melt season (big light grey dot of Fig. 8b). The interplay between these two relations leads to a complex relationship for the mean velocity of the glacier as seen on Fig. 8c. In our experiments, and if the intensity of the melt season is fixed, an



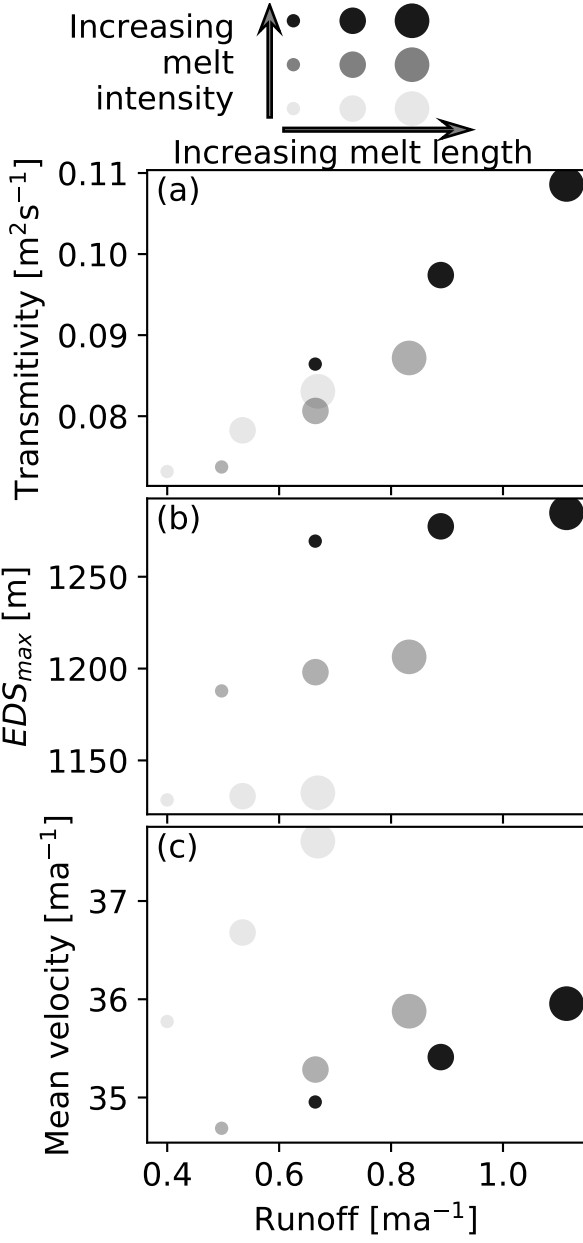

**Figure 8.** Evolution of the subglacial drainage transmitivity (a), of the maximum elevation at which the efficient drainage system is active ($EDS_{max}$,b) and of the mean glacier velocity (c) function of the mean glacier runoff. The different shades of grey represent different melt intensity with the low intensity being lighter than the high intensity. The size of the marker represent the length of the melt season with bigger markers corresponding to longer melt seasons.





increase in runoff and so a longer melt season will lead to an increase in the glaciers velocity. However, for a fixed season length an increase in the melt intensity first drives a sharp decrease in velocity followed by a slow velocity increase. Focusing again on a constant runoff, the velocities in our model are increasing if the length of the melt season increases. That is the reverse scenario of the one of $EDS_{max}$ which was expected. Indeed, a more widespread efficient drainage system allows to drain away a larger amount of water and in the end allows the velocities during summer to settle at a lower point than on a

glacier with a lower effective pressure.

### 3.4   Initiation Length Forcing

The definition of our forcing means that a change in amplitude would lead to small variations in the steepness of the initiation of the melt season. In order to evaluate the effect of this change in the recharge increase, here we present a set of experiments where the length of the initiation has been changed which leads to initiations with different rates. Figure 9 shows the evolution

for simulations in which the initiation rate is altered from the common reference. These simulations have a common runoff which means that the simulation set with the shorter initiation has the lowest melt intensity.

At low elevations, the simulations with the steeper initiation show a very sharp and short-lived spring speed-up Fig. 9a. That contrasts with the simulation with a more gentle initiation where the spring speed-up reaches lower peak velocities and has a longer duration with a secondary peak later in spring. Figure 9a also shows that with a steeper initiation the velocities after the

initial speed-up are relatively slow which contrasts with the more gentle speed up where a number of the ensemble members show quite strong velocities throughout the melt season. These two contrasting responses can be explained by the development speed of the efficient drainage system as seen in Fig. 9e. The simulations with steep initiation periods show the fast development of a very efficient drainage system, whereas the simulations with longer initiation periods show a later development of a less efficient drainage system. This difference explains why the effective pressure in the case of the steep initiation simulation shows

a very steep rebound to a rather high summer value for the effective pressure, while the effective pressure for a more gentle melt season initiation stays rather low throughout the summer, which drives the observed fast velocities. The development of a rather inefficient efficient draining system in the case of the gentle initiation also leads to a large migration upstream of this system and its rather low efficiency produces rather low effective pressures on the higher part of the glacier.

We see on Table 6 that the effect of a change in the initiation length only impacts significantly the lower part of the domain

where the efficient drainage system controls the effective pressure response. The domain means show that the gentler initiation has a larger impact on the velocities with an acceleration of roughly 15% where the steeper initiation only drives a 6% decrease in ice velocity.

It is interesting here to compare these results with the ones from the preceding experiment Fig (7c. In Fig. 9c the simulation with the lowest intensity (sharper initiation) shows a behaviour that is closer to the one of the highest intensity (which is also

the one with the sharper initiation) of Fig. 7c, where the drop in effective pressure is quickly followed by a fast rebound to rather high summer velocities.





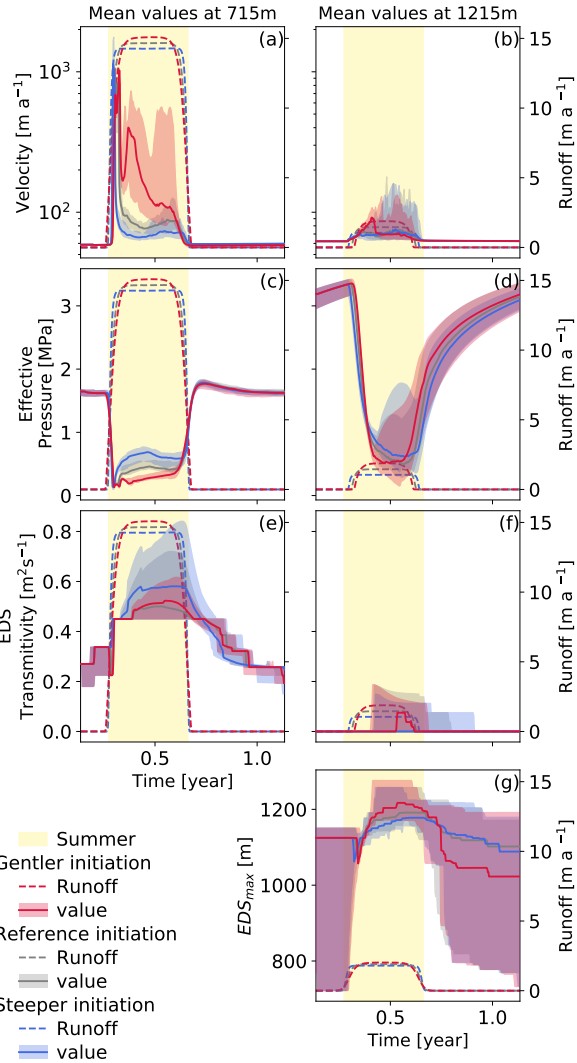

**Figure 9.** Evolution of the velocity (a-b), effective pressure (c-d) and efficient drainage system transmitivity (e-f) at two different altitudes. Evolution of the maximum height of the active EDS (g). Every panel shows the reference simulation in grey, the longer initiation period in red and the shorter initiation period in blue. The same colour scheme is used for the runoffs which are shown as dashed lines and are referenced on the right axis. The yellow shading correspond to the fixed duration summer as defined in Sect. 2.3

This reinforces the hypothesis that the rate of recharge of the subglacial drainage system is might have a similar or even a larger impact than the volume of water that is actually injected into the system (Bartholomaus et al., 2008; Hoffman et al., 2011; Bartholomew et al., 2012; Cowton et al., 2016).





**Table 6.** Velocity difference to the reference simulation for initiation period length. The values in bold fonts are the simulations for which the difference with respect to the reference simulation is significant as per a Wilcoxon test with p=0.01. The summer and winter are defined as fixed periods with summer starting on day 100 and ending on day 241 while winter covers the rest of the year.

|  |  | Annual mean | Summer Mean | Winter Mean |
|---|---|---|---|---|
| Steeper initiation $SSU_{max}$ = 997 m $EDS_{max}$ = 1174 m | domain | **-6.60%** | **-12.62%** | **0.09%** |
|  | 715 m | **-12.06%** | **-22.13%** | **0.44%** |
|  | 965 m | **-2.13%** | **-4.76%** | **0.10%** |
|  | 1215 m | -0.45% | -1.17% | **0.08%** |
|  | 1465 m | -0.02% | -0.03% | **-0.01%** |
| Gentler initiation $SSU_{max}$ = 1037 m $EDS_{max}$ = 1201 m | domain | **23.86%** | **45.82%** | **-0.30%** |
|  | 715 m | **30.24%** | **56.48%** | **-2.10%** |
|  | 965 m | **6.79%** | **14.78%** | **0.23%** |
|  | 1215 m | 0.37% | 0.64% | **0.21%** |
|  | 1465 m | **0.02%** | 0.02% | **0.02%** |

## 3.5 Shortcomings

As with any modelling study, the results presented here might be impacted by the model design and the experiment set-up. It is important to note that subglacial hydrological models have not converged on a standard way to treat the subglacial drainage system yet and the SHMIP exercise (de Fleurian et al., 2018) has shown that there is some discrepancies between the different approaches. However, we think that the results presented here are robust in this model and have a physical explanation which is coherent with existing theories and observations of subglacial drainage.

One specific shortcoming in our experiments is that the recharge of the subglacial drainage system has the same distribution and timing as the surface runoff. This is not what is expected in a natural setting where water produced at the surface will transit at the surface of the ice for a given time before going through the ice through moulins and entering the subglacial drainage system through these localised injection points. Scholzen et al. (2021) showed a limited impact on the subglacial drainage system between simulations lead with a spatially homogeneous or discrete basal recharge. Their results show a difference in the timing of the effective pressure response to the runoff but only slight variations in amplitude or length of the pressure pulse. This is coherent with the results of the SHMIP intercomparison (de Fleurian et al., 2018), it seems that the moulin location does not have a large impact on the overall evolution of the subglacial water pressure. It seems however that the highest injection point on the domain could alter the model results, in this study this highest injection point is defined as the highest point where melt exists on the glacier. However, if the supra and intraglacial drainage was consider it is most likely that the highest injection point would not change on a yearly basis to adapt to the changes in the melt season. Gagliardini and Werder (2018) has modelled moulin migration rates from roughly one to ten meters of elevation per year depending on the setting.





## 4 Discussion

The results of our model suggest that the relationship between ice velocity and meltwater runoff is also strongly influenced by
the distribution of the melt during the year. Hence if an increase in runoff is driven by a longer melt season the mean annual
velocities of the glaciers will show a strong increase at all elevations. However, an increase in the intensity of the melt season
first drives a reduction of the ice velocities before they start to increase again with what seems to be a smaller rate. The mean
annual velocities can however not explain the complexity of the lubrication feedback. The effect of a change in the length of the
melt season has a similar effect on all regions of the glacier, and the velocity increases linearly with the length of the melt season
(Table 4). This differs from the response to an increase in the intensity of the melt season. There the response is different at low
elevations, where the subglacial drainage is controlled by the efficient component, and at high elevations, where the drainage is
controlled by the inefficient components. At higher elevations, our results compare well with those of Gagliardini and Werder
(2018) where we see that an increase of the recharge in the regions controlled by the inefficient drainage system leads to an
acceleration of the glacier. However, at lower elevations our results diverge from preceding studies. In the lower parts of the
glacier, where the subglacial drainage is controlled by an efficient drainage system, an increase in the intensity of the melt
season leads to a decrease in the ice-flow velocities. We explain this result through the evolution of the efficient drainage
system in those simulations (Fig. 7e-g). In these figures we can see a faster development of the efficient drainage system
to higher elevations for the more intense melt season. The development of this system leads to an increase in the effective
pressure on the lower part of the glacier, and so keeps the velocities at levels that are comparable to the ones of our reference
simulations. In the case of the low intensity melt season however, the water recharge is not sufficient to trigger the development
of a well-developed efficient drainage system, that leads to low effective pressures throughout the melt season which in turn
induce a large increase in velocities compared to our reference simulation. It must be noted however that these conclusions
only hold on seasonal velocities and that a higher intensity runoff will lead to higher maximum velocities. This result can be
compared to the observations made on lake drainage by Tedesco et al. (2013b) where a fast draining lake triggered a large
and short lived speed-up while a slower draining lake only generated a mild speed-up after which the velocities stabilised at a
higher level than what was recorded before the lake drainage. We can see the effect of the rate of recharge on the experiments
where the length of the initiation of the melt season was changed (Fig. 9). In this case, the variation in the duration of the
initiation of the melt season leads to quite large variations in the overall velocities of the glacier. Here, the sharper initiation
leads to slower velocities, but this simulation also has a slightly smaller amplitude to keep the overall runoff identical for all
simulations. This is in contradiction with the the preceding experiments where the smaller amplitudes were driving a faster
glacier. But the similarities here lie in the sharpness of the runoff curve at the begining of the melt season. Hence a sharp rise of
the temperatures at the begining of the melt season has the potential to produce a large amount of meltwater which in turn could
trigger an early activation of the subglacial drainage system and lead to gentler velocities throughout the summer (after a large
spring speed-up event). This large impact of the slope of the temperature rise at the begining of the melt season is problematic
to provide estimates of the impact of the lubrication feedback as this parameter is highly variable and complex to characterise
in the existing dataset.



In our model, the observed mean velocities are mainly driven by the lower regions of the glaciers where the velocities are significantly higher. That is clear when comparing the velocity evolution at different altitudes to the domain mean velocity on Fig. 3. This explains why in our model the mean velocities are largely influenced by the activation of the efficient drainage

system which in turn is tightly linked to the intensity of the melt season. These results could be different with the use of another subglacial hydrology model producing a different response, specifically at higher elevations as can be seen in the SHMIP intercomparison exercise (de Fleurian et al., 2018).

The large difference in behaviour between the lower and higher part of the glacier make it challenging to extrapolate the results to a longer term evolution of the ice dynamics without performing the actual simulations. The two different behaviours

are tightly linked to the region in which the efficient drainage system develops which itself is quite sensitive to the intensity of the melt season as seen on Fig. 8b. An increase in the length of the melt season should not alter in a large way the spread of the efficient drainage. Moreover, the variation in melt season length is affecting the whole glacier in the same way so that the response here is quite clear and longer melt season should lead to overall faster glaciers. However, it is not expected that the current evolution in climate would only alter the length of the melt season in Greenland and our model shows that the impact

of lengthening the melt season is actually only one third of the acceleration that we observe when we reduce the temperature by a comparable amount. This shows that at least in our model, the effect of the intensity of the melt season is more marked than its length. However, this second effect as larger implications on the migration of the efficient drainage system upstream. In our experiments, an increase in runoff amplitude leads to a slight slow down of the glacier which is linked to the larger spread of the efficient drainage system. The large runoff at higher altitudes also drives a faster glacier there and in the long term

that could induce a larger region of faster flow which would lead to faster velocities overall. On the other hand, a longer melt season of lower intensity drives a faster glacier in the marginal region, but also shows a decrease in the upstream velocities. This should however be taken with caution as a lowering of the glacier surface would lead to an intensification of the melt which we have shown to be a potential negative feedback on ice velocities. To better understand the impact of these processes on future Greenland Ice Sheet velocities, studies could focus on finding out the main expected trends in length and intensity of

future melt seasons.

These different scenarios, with counter-intuitive results for the seasonal velocities, show that the full subglacial drainage system, and particularly its efficient component, must be included in studies that aim to quantify the effect of meltwater lubrication feedback.

## 5 Conclusions

We developed a set of experiments that allows the comparison of the effect of different parameters impacting the distribution of runoff throughout the melt season. The use of forcing scenarios based on the ERA5 reanalysis dataset gives us confidence that the variations in intensity and length of the melt season that we tested here are representative of the range of existing melt seasons. Our results show that under a constant runoff, an increase in the length of the melt season drives an overall faster glacier. With simulations spanning a range of different runoff intensities we show that an increase in melt intensity





originally leads to a higher maximum velocity at low elevations but then translates to a lower mean summer velocity over the whole domain and an overall slowdown of the glacier. This behaviour is mostly due to the development of a very transmissive efficient drainage system at low elevations which keeps the effective pressures at a high level throughout the melt season, and so leads to relatively low summer velocities when averaged over the whole domain. This study can not give a definite answer on the impact of an increase in runoff on the long term evolution of glacier velocities. Indeed, the impact of the melt

season amplitude is radically different at different altitudes and these regions are delimited by the existence or not of an efficient drainage system. Our experiments show that an increase in the amplitude of the melt season leads to a larger extent of the region where the drainage is controlled by the efficient subglacial drainage system. This change of regime on the highest regions of the glaciers might significantly alter the velocity profile of the glacier with regions at higher altitudes then experiencing spring speed-up events. This finding emphasises the fact that subglacial drainage models with efficient drainage components should

be used if one wants to give an accurate assessment of the effect of meltwater on the overall dynamics of the Greenland Ice sheet.

*Code and data availability.* The Ice-sheet and Sea-level System Model is freely available at https://issm.jpl.nasa.gov/. The model outputs coresponding to this study are available on zenodo https://doi.org/10.5281/zenodo.5959181

*Author contributions.* Experiment design was collectively done by all co-authors. RD designed the forcing. BdF ran the experiments and

wrote the manuscript with input from all co-authors.

*Competing interests.* The authors declare that they have no conflict of interest.

*Acknowledgements.* This work is part of the SWItchDyn project funded by the Research Council of Norway (NFR-287206) and the Bjerknes Centre for Climate Research strategic project RISES. Computing was performed on the resources provided by UNINETT Sigma2 – the National Infrastructure for High Performance Computing and Data Storage in Norway (NN9635K and NS9635K).



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
