# Peer review of "Impact of runoff temporal distribution on ice dynamics."

_The Cryosphere, 2022_

## Referee Comment (RC1)

**Review of "Impact of runoff temporal distribution on ice dynamics", by de Fleurian et al.**

**Summary:**

This paper explores how the meltwater-lubrication feedback might be impacted by varying intensity and duration of the melt season for the Greenland Ice Sheet. The authors use a coupled ice flow and basal hydrology model in an idealized domain with simplified forcing to answer this question. They test the response of the ice sheet flow rate to changes in the intensity of the melt season, changes in the duration of the melt season, and changes in the rapidity of the onset of the melt season. They find a contrasting response, such that increasing intensity of the melt season induces a net slowdown (through the development of a more efficient drainage system), while increasing duration of the melt season induces a net speedup. They also discuss differences in the ice dynamic response at different altitudes.

**Overall Review:**

The question of the meltwater-lubrication feedback for Greenland is a long-standing question in efforts to predict sea level rise in a warming climate. There has long been uncertainty about whether the increasing altitude extent of melting in a warming climate, which tends to increase the area of the ice base receiving surface melt input, will produce a positive feedback, or whether the increasing efficiency of the subglacial hydrological system at lower altitudes will produce a negative feedback.

This paper makes an important contribution to that literature. The ice flow and basal hydrology models that the authors employ are well suited to the problem at hand. The experimental design is well formulated, the analysis is thorough, and the presentation is sound. The biggest gap in this paper, in my opinion, is that they do not explore the question of short-term temporal variability in the melt input, although they bump up against the edge of that topic in their discussion of the importance of the rate of change of the melt forcing in the experiment that varied the onset of the melt season. However, I do not consider that an obstacle to publication. It is not the responsibility of the authors to answer every potential question about their topic. They set themselves a well-defined question (exploring the effect of changes in intensity and duration in the melt season on ice flow rates) and they designed an experimental setup that is well-suited to answering that question. The issue of short-term variability in melt input can be addressed with a bit more discussion and perhaps a call for future work. I also have numerous other comments and suggestions for this paper, but none of them rises to the level of a major issue that should impede publication.

Overall, my recommendation is to publish with minor revisions.

**Detailed Comments:**

L1-2: The first sentence of the abstract is a bit ambiguous. On a first reading, it sounds like you're saying that records (ie, observations) of meltwater production have a surprisingly high recurrence, but I think what you mean to say is that *record-highs* of meltwater production have a surprisingly high recurrence. Maybe change the first sentence to, "Record-highs of meltwater production…"

L11: "Furthermore…."
It would probably be better to start this sentence with "However…", since the message of this sentence is somewhat contradictory to the previous one (in the previous sentence we learn that longer melt seasons cause the glacier to speed up, but here we learn that more intense melt seasons cause the glacier to slow down, setting up a tension between "more melt" expressed through the length of the melt season and "more melt" expressed through the intensity of the melt season).

L20: "..identified in southwest Greenland that a shift in the runoff regime took place in 2003…"
Rearrange to: "...identified that a shift in the runoff regime in southwest Greenland took place in 2003…"

L21: compare → compared

L24: "the length of the melt season has been increasing" → "the length of the melt season has also been increasing"
Continuing on our theme of emphasizing the contrast between length and intensity of the melt season.

L33-34: the "e.g." should be at the beginning of the citation

L34: smaller pressure → lower pressure

L36: add a comma after "available"

L40: "will allow to drain the provided water" → "will allow the provided water to drain"

L42: "seemingly opposing results" → "seemingly opposed results"

L42-46: This entire paragraph should be one sentence, separated into clauses by the colon and a subsequent comma. The way to organize a list like this is: "There are two effects of the thing we are talking about: (i) blah blah blah, and, (ii) bleh bleh bleh." In addition, this sentence should make clear that the reason for the difference in behavior is that the high-elevation regions start from a different baseline state than the low-elevation regions, so they are on different sides of the tipping point discussed in the previous paragraph. A possible way to reword this paragraph could then be: "This threshold behaviour leads to seemingly opposed results of an increase in meltwater availability that can be observed in western Greenland: (i) at high elevations, the subglacial hydrologic system begins in an inefficient state, and thus increases in water supply will increase the subglacial water pressure and lead to faster glaciers (e.g. Zwally et al., 2002; Doyle et al., 2014), but, (ii) at lower elevations, the increased water supply will only increase the efficiency of the drainage system, leading to lower water pressure and a slower ice flow (e.g. Sundal et al., 2011; Sole et al., 2013; Tedstone et al., 2015)."

L47: no need for the comma after "observations"

L48: forgot the space in "subglacial drainage"

L53: "has the recent study" → "as the recent study"

L60: replace the semicolon separating the citations with "and"

L61: "We will first give an overview of the component of the model which are specific to this study"
Is there only one component of the model that is specific to this study? If so, then the sentence should read, "...the component of the model that is…" If not, then it should read, "...the components of the model which are…"

L61-62: It's a bit weird to have two sentences start with "we first…" You can only do one thing first! Maybe start the second sentence, "We then present the results…"

Eqn1: Good choice of sliding law! It is much better to have a plastic or pseudo-plastic bed for this sort of study than a Weertman or Budd law.

L75-80: I assume that "n" represents the rheological exponent for ice, but you should still specify the meaning of all variables used in your equations.

L86-88: The explanation of the meaning of eqn. 2 is a little unclear. You start the explanation saying, "This equation involves…" but then only discuss the first term; then you say, "The other term represents…" but it was not clear that the previous sentence was only discussing the first term. Also, you should mention that the first term only includes melting due to viscous dissipation within the hydrological system, not other heat sources and sinks. Maybe rephrase as: "The first term in this equation represents the growth of the efficient system by the melting of ice walls through the heat generated by dissipation, where...[list variable meanings here]. The second term represents the closing of the efficient system by ice creep, where...[variable definitions]."

General model description: Equation 2 only includes melt from viscous dissipation within the hydrological system, not from any other source. Nor does equation 2 include a mass conservation for the water system. However, we can infer that additional melt sources are possible because surface melt input is routed to the bed, and presumably mass conservation is handled through the inefficient part of the double continuum model. I understand that these issues are addressed in the cited references, but it would be good to include a bit more information in this paper as well. In particular, I would like to know what other sources of melt input are considered: in addition to viscous dissipation in the water system and surface melt draining to the bed, does the model also consider melt from shear heating as the ice slides over the base? What about melt from geothermal heating? I think that including slightly more information about how the model works would help this part of the paper.

L93-101: This seems like a reasonable coupling architecture.

L105: You should probably mention here that you chose a flat bed at z=465 m instead of a bed at z=0 m in order to facilitate comparisons with southwestern Greenland. Otherwise, this number seems a bit random.

L161: "...test if the perturbation lead to…" → "...test if the perturbation leads to…"

L163: If the probability that the medians are different is 1%, doesn't that mean that the confidence level is 99%?

Figure 2: Panels a-c represent the spatial mean over the whole domain, right? The text (L166) implies that that is what you are showing, but it would be helpful to state that in the caption as well.

L196-197: "Even higher up on the glacier the effective pressure is driven by downstream activity as there is no runoff at these elevations."
Question: do you include a background level of subglacial melt input to the hydrological system, so that there is actually a small source of water high up on the glacier? Or is water flowing upstream to get into these regions? Because if water is indeed flowing upstream, then that seems a little unrealistic. This is where my previous request for more information about the model becomes relevant.

L214-215: "Since we chose to keep the runoff constant for this set of simulations, changes in the length of the melt season simultaneously impact the melt intensity."

Maybe it would be better to phrase this as, "For this set of simulations, we wish to investigate the affect of changes in the melt season length, independent of changes in the integrated melt volume, so we vary the melt intensity inversely with the melt season length." Rephrasing it this way puts the emphasis on the reasoning behind your experimental choice.

While you are, of course, free to devise any experimental design you wish for an idealized model, it might also be worth pointing out that these sorts of compensating changes (where melt season length and intensity trade off with one another to keep the integrated melt roughly constant) are not likely to happen in reality. In reality, it is more likely that a warming climate will produce increases of both the intensity and the duration of the melt season. However, this set of experiments does nicely compliment the experiments shown in section 3.3, allowing you to separate out the effect of melt season length while keeping integrated melt constant.

L229-230: "While the reference simulation was only showing..." → "While the reference simulation only showed..."

L230: "...there is a quite large acceleration…" → "...there is quite a large acceleration…"

L239: "This contrasts with the short melt season…"
Do you mean that it contrasts with the long season, since the previous sentence was discussing the short season?

L247: "contrasted" → "contrasting"

L269-270: Rephrase this sentence in the active voice. Maybe something like, "In order to discriminate between the effects of melt season intensity and length, we release the requirement that the runoff must be equal in all simulations."

L283-284: "However, the extreme values for the longer melt seasons tend to show more important acceleration events happening at the end of the melt season."
What do you mean by "extreme values"? Are you referring to individual ensemble members as opposed to the ensemble median for each simulation? If so, maybe consider rephrasing to, "However, while the median summer velocity is similar for all simulations, individual ensemble members with large acceleration events late in the melt season are more common in the longer melt season."

L290: "Comparing the simulations with different intensities yields more significant differences between simulations"
Actually, by comparing tables 4 and 5, it looks like the simulations with different intensities actually have *fewer* significant differences, but those differences are larger in amplitude. Maybe it would be best to replace "more significant differences" with "larger differences", to avoid confusion between 'significant' meaning 'big or important' and 'significant' meaning statistical significance.

L349: "We see on Table 6…" → "We see in Table 6..."

L357: "...is might have…" → "...might have…."

L363: "...is some discrepancies…" → "...are some discrepancies…"

Section 3.5: Shortcomings
        This section looks like it would go better in the discussion than in the results. In addition, while you discussed the lack of spatial heterogeneity in meltwater injection in your model, you have not mentioned the lack of temporal heterogeneity. As you mentioned immediately before this

section, there is a body of work suggesting that the rate of change of subglacial water input may be more important than the actual volume of input.  In that case, high-frequency temporal variability in the meltwater input (from both the daily cycle and from synoptic weather variability) might play an important role in governing the response of the subglacial hydrological system.  In a warming climate, we would expect not only an increase in melt season intensity and duration, but also an increase in synoptic melt variability, including an increase in short-duration melt extremes like the examples cited as motivation in the introduction of this paper.

Overall, it is fine that you have chosen the particular experimental design that you did, as your experiments are well suited to answering the question of length vs intensity of the melt season. However, when we move from these simplified idealized setups and start to think about the implications of your results for the future evolution of the Greenland Ice Sheet, the largest missing piece of the puzzle is, in my opinion, the lack of short-term temporal variability in your melt input. I think that it is important to discuss the potential role of short-term melt variability in the discussion section.

L397-411:  Discussion of recharge rates.
This discussion touches on the issue I mentioned above, the importance of the rate of change of melt input.  However, the importance of the rate of change means that not only is the onset of the melt season important, but so is synoptic temporal variability throughout the melt season.  This would be a good place to include a few sentences about short-term temporal variability. Furthermore, as a matter of presentation it might be good to emphasize this topic by giving it its own paragraph.  The first paragraph of the discussion section is too long anyway, so consider adding a paragraph break somewhere around lines 395-400.

L409-411:  "This large impact of the slope of the temperature rise at the begining of the melt season is problematic to provide estimates of the impact of the lubrication feedback as this parameter is highly variable and complex to characterise in the existing dataset."
This sentence is difficult to parse.  Consider rephrasing to, "This large impact of the slope of the temperature rise at the beginning of the melt season is problematic for efforts to estimate the lubrication feedback, because this parameter is highly variable and complex to characterize in the existing dataset."  I would also add that the large impact of the slope of the temperature rise at the beginning of the melt season also reinforces the argument I made above that variability in melt rate during the melt season may also have a big influence on the hydrological and ice dynamic response.

L412-412:  "In our model, the observed mean velocities are mainly driven by the lower regions of the glaciers where the velocities are significantly higher."
Hmmm….Does this mean that you might have more representative metrics of ice dynamics if you computed relative speed-up instead of absolute speed-up?  Would it be too much work to add relative speed-up to your analysis?

L423-427:  "However, it is not expected that the current evolution in climate would only alter the length of the melt season in Greenland and our model shows that the impact of lengthening the melt season is actually only one third of the acceleration that we observe when we reduce the temperature by a comparable amount. This shows that at least in our model, the effect of the intensity of the melt season is more marked than its length."
However, when you *increased* the temperature by a comparable amount, you saw no significant change in the mean annual velocity of the glacier (Table 5).  The response to melt season intensity was asymmetric.  Since we expect both intensity and length of the melt season to increase in a warming climate, this suggests that your model actually supports the opposite conclusion:  in a warming climate, the speedup caused by an increase in melt season length is likely to outweigh the (statistically insignificant) slowdown caused by an increase in melt season intensity.

L427: "as larger implications" → "has larger implications"

---

## Author Comment (AC1)

**Answer to the reviewers on our manuscript entitled : "Impact of runoff temporal distribution on ice dynamics."**

Basile de Fleurian, Richard Davy and Petra M. Langebroek

We want to thank both reviewer for their comments and observations that will certainly increase the quality of our manuscript. Find bellow our answer to the comments with our answers highlighted in blue.

**1 Answer to Michael Wolovick (Reviewer #1):**

**1.1 Overall comments**

The question of the meltwater-lubrication feedback for Greenland is a long-standing question in efforts to predict sea level rise in a warming climate. There has long been uncertainty about whether the increasing altitude extent of melting in a warming climate, which tends to increase the area of the ice base receiving surface melt input, will produce a positive feedback, or whether the increasing efficiency of the subglacial hydrological system at lower altitudes will produce a negative feedback. This paper makes an important contribution to that literature. The ice flow and basal hydrology models that the authors employ are well suited to the problem at hand. The experimental design is well formulated, the analysis is thorough, and the presentation is sound. The biggest gap in this paper, in my opinion, is that they do not explore the question of short-term temporal variability in the melt input, although they bump up against the edge of that topic in their discussion of the importance of the rate of change of the melt forcing in the experiment that varied the onset of the melt season. However, I do not consider that an obstacle to publication. It is not the responsibility of the authors to answer every potential question about their topic. They set themselves a well-defined question (exploring the effect of changes in intensity and duration in the melt season on ice flow rates) and they designed an experimental setup that is well-suited to answering that question. The issue of short-term variability in melt input can be addressed with a bit more discussion and perhaps a call for future work.

We considered at some point to add those considerations into the paper seeing the large impact that a small change in temperature increase slope has on the response of the model. However that would have required a brand new set of experiments and would have easily doubled the size of the present manuscript. We added a few comments on this point and point towards the needs of this kinds of study in the future.

I also have numerous other comments and suggestions for this paper, but none of them rises to the level of a major issue that should impede publication. Overall, my recommendation is to publish with minor revisions.

**1.2   Detailed comments**

- L1-2: The first sentence of the abstract is a bit ambiguous. On a first reading, it sounds like you're saying that records (ie, observations) of meltwater production have a surprisingly high recurrence, but I think what you mean to say is that record-highs of meltwater production have a surprisingly high recurrence. Maybe change the first sentence to, "Record-highs of meltwater production..."

  Your assumption is right, this will be reformulated in the manuscript

- L11: "Furthermore..." It would probably be better to start this sentence with "However...", since the message of this sentence is somewhat contradictory to the previous one (in the previous sentence we learn that longer melt seasons cause the glacier to speed up, but here we learn that more intense melt seasons cause the glacier to slow down, setting up a tension between "more melt" expressed through the length of the melt season and "more melt" expressed through the intensity of the melt season).

  Agreed, this has been changed

- L20: "...identified in southwest Greenland that a shift in the runoff regime took place in 2003..." Rearrange to: "...identified that a shift in the runoff regime in southwest Greenland took place in 2003..."

  Changed

- L21: compare → compared

  Changed

- L24: "the length of the melt season has been increasing" → "the length of the melt season has also been increasing" Continuing on our theme of emphasizing the contrast between length and intensity of the melt season.

  Changed

- L33-34: the "e.g." should be at the beginning of the citation

  Indeed, that as been corrected

- L34: smaller pressure → lower pressure

  Changed

- L36: add a comma after "available"

  Added

- L40: "will allow to drain the provided water" → "will allow the provided water to drain"

  Changed

- L42: "seemingly opposing results" → "seemingly opposed results"

  Changed

- L42-46: This entire paragraph should be one sentence, separated into clauses by the colon and a subsequent comma. The way to organize a list like this is: "There are two effects of the thing we are talking about: (i) blah blah blah, and, (ii) bleh bleh bleh." In addition, this sentence should make clear that the reason for the difference in behavior is that the high-elevation regions start from a different baseline state than the low-elevation regions, so they are on different sides of the tipping point discussed in the previous paragraph. A possible way to reword this paragraph could then be: "This threshold behaviour leads to seemingly opposed results of an increase in meltwater availability that can be observed in western Greenland: (i) at high elevations, the subglacial hydrologic system begins in an inefficient state, and thus increases in water supply will increase the subglacial water pressure and lead to faster glaciers (e.g. Zwally et al., 2002; Doyle et al., 2014), but, (ii) at lower elevations, the increased water supply will only increase the efficiency of the drainage system, leading to lower water pressure and a slower ice flow (e.g. Sundal et al., 2011; Sole et al., 2013; Tedstone et al., 2015)."

  This paragraph has been rephrased.

- L47: no need for the comma after "observations"

  Removed

- L48: forgot the space in "subglacial drainage

  Corrected

- L53: "has the recent study" → "as the recent study"

  Corrected

- L60: replace the semicolon separating the citations with "and"

  This is standard formatting and have been left as is

- L61: "We will first give an overview of the component of the model which are specific to this study" Is there only one component of the model that is specific to this study? If so, then the sentence should read, "...the component of the model that is..." If not, then it should read, "...the components of the model which are..."

  Corrected

- L61-62: It's a bit weird to have two sentences start with "we first..." You can only do one thing first! Maybe start the second sentence, "We then present the results..."

  This has been rephrased

- Eqn1: Good choice of sliding law! It is much better to have a plastic or pseudo-plastic bed for this sort of study than a Weertman or Budd law.

  This is our opinion too

- L75-80: I assume that "n" represents the rheological exponent for ice, but you should still specify the meaning of all variables used in your equations.

  You are right as shown in Table 1 the description has been added here too

- L86-88: The explanation of the meaning of eqn. 2 is a little unclear. You start the explanation saying, "This equation involves..." but then only discuss the first term; then you say, "The other term represents..." but it was not clear that the previous sentence was only discussing the first term. Also, you should mention that the first term only includes melting due to viscous dissipation within the hydrological system, not other heat sources and sinks. Maybe rephrase as: "The first term in this equation represents the growth of the efficient system by the melting of ice walls through the heat generated by dissipation, where...[list variable meanings here]. The second term represents the closing of the efficient system by ice creep, where...[variable definitions]."

  Rephrased

- General model description: Equation 2 only includes melt from viscous dissipation within the hydrological system, not from any other source. Nor does equation 2 include a mass conservation for the water system. However, we can infer that additional melt sources are possible because surface melt input is routed to the bed, and presumably mass conservation is handled through the inefficient part of the double continuum model. I understand that these issues are addressed in the cited references, but it would be good to include a bit more information in this paper as well. In particular, I would like to know what other sources of melt input are considered: in addition to viscous dissipation in the water system and surface melt draining to the bed, does the model also consider melt from shear heating as the ice slides over the base? What about melt from geothermal heating? I think that

including slightly more information about how the model works would help this part of the paper.

Some more details have been added on the model design

- L93-101: This seems like a reasonable coupling architecture.

- L105: You should probably mention here that you chose a flat bed at z=465 m instead of a bed at z=0 m in order to facilitate comparisons with southwestern Greenland. Otherwise, this number seems a bit random.

  A sentence have been added for clarification

- L161: "...test if the perturbation lead to..." → "...test if the perturbation leads to..."

  Changed to plural for perturbation instead

- L163: If the probability that the medians are different is 1%, doesn't that mean that the confidence level is 99%?

  Yes that is right and is corrected in the manuscript

- Figure 2: Panels a-c represent the spatial mean over the whole domain, right? The text (L166) implies that that is what you are showing, but it would be helpful to state that in the caption as well.

  The caption was stating it already but not in a very clear way, this has been modified.

- L196-197: "Even higher up on the glacier the effective pressure is driven by down-stream activity as there is no runoff at these elevations." Question: do you include a background level of subglacial melt input to the hydrological system, so that there is actually a small source of water high up on the glacier? Or is water flowing upstream to get into these regions? Because if water is indeed flowing upstream, then that seems a little unrealistic. This is where my previous request for more information about the model becomes relevant.

  Yes there is indeed a background geothermal heat flux related water input.

- L214-215: "Since we chose to keep the runoff constant for this set of simulations, changes in the length of the melt season simultaneously impact the melt intensity." Maybe it would be better to phrase this as, "For this set of simulations, we wish to investigate the affect of changes in the melt season length, independent of changes in the integrated melt volume, so we vary the melt intensity inversely with the melt season length." Rephrasing it this way puts the emphasis on the reasoning behind your experimental choice. While you are, of course, free to devise any experimental design you wish for an idealized model, it might also be worth pointing out that these sorts of compensating changes (where melt season length and intensity trade off with

one another to keep the integrated melt roughly constant) are not likely to happen in reality. In reality, it is more likely that a warming climate will produce increases of both the intensity and the duration of the melt season. However, this set of experiments does nicely compliment the experiments shown in section 3.3, allowing you to separate out the effect of melt season length while keeping integrated melt constant.

We rephrased the sentence has suggested. No more details were given here on the reality of the set-up here as we feel that his has been sufficiently described in the discussion section.

- L229-230: "While the reference simulation was only showing..." → "While the reference simulation only showed..."

  Changed

- L230: "...there is a quite large acceleration..." → "...there is quite a large acceleration..."

  Changed

- L239: "This contrasts with the short melt season..." Do you mean that it contrasts with the long season, since the previous sentence was discussing the short season?

  Yes, this has been corrected

- L247: "contrasted" → "contrasting"

  Changed

- L269-270: Rephrase this sentence in the active voice. Maybe something like, "In order to discriminate between the effects of melt season intensity and length, we release the requirement that the runoff must be equal in all simulations."

  Rephrased

- L283-284: "However, the extreme values for the longer melt seasons tend to show more important acceleration events happening at the end of the melt season." What do you mean by "extreme values"? Are you referring to individual ensemble members as opposed to the ensemble median for each simulation? If so, maybe consider rephrasing to, "However, while the median summer velocity is similar for all simulations, individual ensemble members with large acceleration events late in the melt season are more common in the longer melt season."

  Yes that was what I was referring to here. The sentence have been rephrased with your suggestion.

- L290: "Comparing the simulations with different intensities yields more significant differences between simulations" Actually, by comparing tables 4 and 5, it looks like the simulations with different intensities actually have fewer significant differences, but those differences are larger in amplitude. Maybe it would be best to replace "more significant differences" with "larger differences", to avoid confusion between "significant" meaning "big or important" and "significant" meaning statistical significance.

  You are completely right and that was an oversight on my side, this is now changed

- L349: "We see on Table 6..." → "We see in Table 6..."

  Changed

- L357: "...is might have..." → "...might have..."

  Corrected

- L363: "...is some discrepancies..." → "...are some discrepancies..."

  Corrected

- Section 3.5: Shortcomings This section looks like it would go better in the discussion than in the results. In addition, while you discussed the lack of spatial heterogeneity in meltwater injection in your model, you have not mentioned the lack of temporal heterogeneity. As you mentioned immediately before this section, there is a body of work suggesting that the rate of change of subglacial water input may be more important than the actual volume of input. In that case, high-frequency temporal variability in the meltwater input (from both the daily cycle and from synoptic weather variability) might play an important role in governing the response of the subglacial hydrological system. In a warming climate, we would expect not only an increase in melt season intensity and duration, but also an increase in synoptic melt variability, including an increase in short-duration melt extremes like the examples cited as motivation in the introduction of this paper. Overall, it is fine that you have chosen the particular experimental design that you did, as your experiments are well suited to answering the question of length vs intensity of the melt season. However, when we move from these simplified idealized setups and start to think about the implications of your results for the future evolution of the Greenland Ice Sheet, the largest missing piece of the puzzle is, in my opinion, the lack of short-term temporal variability in your melt input. I think that it is important to discuss the potential role of short-term melt variability in the discussion section.

  This section have been reworked and introduced in different locations of the discussion section. Regarding the temporal variability of the input we added a few sentences in the discussion section and pointed to this specific point as an important step forward for further studies.

- L397-411: Discussion of recharge rates. This discussion touches on the issue I mentioned above, the importance of the rate of change of melt input. However, the importance of the rate of change means that not only is the onset of the melt season important, but so is synoptic temporal variability throughout the melt season. This would be a good place to include a few sentences about short-term temporal variability. Furthermore, as a matter of presentation it might be good to emphasize this topic by giving it its own paragraph. The first paragraph of the discussion section is too long anyway, so consider adding a paragraph break somewhere around lines 395-400.

  We modified the structure of the discussion section and added a sentence on the importance of short term varaition. We do not want however to go to much in the detail of this problem as we think that it would overload an already quite dense manuscript.

- L409-411: "This large impact of the slope of the temperature rise at the begining of the melt season is problematic to provide estimates of the impact of the lubrication feedback as this parameter is highly variable and complex to characterise in the existing dataset." This sentence is difficult to parse. Consider rephrasing to, "This large impact of the slope of the temperature rise at the beginning of the melt season is problematic for efforts to estimate the lubrication feedback, because this parameter is highly variable and complex to characterize in the existing dataset." I would also add that the large impact of the slope of the temperature rise at the beginning of the melt season also reinforces the argument I made above that variability in melt rate during the melt season may also have a big influence on the hydrological and ice dynamic response.

  We rephrased as : "However, the slope of the temperature rise at the begining of the melt season is higly variable and complex to characterise in the existing dataset which makes it problematic to provide reliable estimates of the impact of this parameter on the lubrication feedback."

- L412-412: "In our model, the observed mean velocities are mainly driven by the lower regions of the glaciers where the velocities are significantly higher." Hmmm... Does this mean that you might have more representative metrics of ice dynamics if you computed relative speed-up instead of absolute speed-up? Would it be too much work to add relative speed-up to your analysis?

  The relative speed-up might be more representative in some case and that is why we elected to present it in the different tables as a velocity difference to the reference simulation. The comment cited here is more related to the velocity evolution as seen on Figure 3 where the evolution of the mean velocity is very similar to the one observed at the lower elevations.

- L423-427: "However, it is not expected that the current evolution in climate would only alter the length of the melt season in Greenland and our model shows that the impact of lengthening the melt season is actually only one third of the acceleration that we observe when we reduce the temperature by a comparable amount. This shows that at least in our model, the effect of the intensity of the melt season is more marked than its length." However, when you increased the temperature by a comparable amount, you saw no significant change in the mean annual velocity of the glacier (Table 5). The response to melt season intensity was asymmetric. Since we expect both intensity and length of the melt season to increase in a warming climate, this suggests that your model actually supports the opposite conclusion: in a warming climate, the speedup caused by an increase in melt season length is likely to outweigh the (statistically insignificant) slowdown caused by an increase in melt season intensity.

  Yes I think that your analyse makes more sens that the one that I wrote down. I was somehow focusing on acceleration rather than increase of both length and intensity of the melt season. This part of the manuscript have been reformulated.

- L427: "as larger implications" → "has larger implications"

  Corrected

**2 Answer to Reviewer #2:**

This study by de Fleurian and colleagues uses a subglacial hydrology model coupled to an ice flow model to test the impact of melt season duration and intensity on ice dynamics, for a Greenland-style idealised land terminating glacier. The presentation of results is very methodical, although quite dense in places. However, I think the wordiness is probably unavoidable to ensure the high level of detail, and the discussion offers a good summary. I only have a few minor comments and recommend that the paper is published after minor revisions.

**2.1 General comments**

The meltwater lubrication feedback is one way that meltwater can impact ice dynamics. I realise the focus here is on land terminating glaciers, but in marine terminating glaciers, the subglacial drainage system has been shown to have an impact on frontal ablation (e.g. see Slater et al., 2015, doi: 10.1002/2014GL062494). Perhaps the authors could comment on this somewhere in the introduction or discussion.

  That is true and something that we have overlooked. We added a sentence to advertise this point in the introduction

The discussion could go further to discuss the potential impact of high frequency variability in melt rate over the season. The fact that the results are so sensitive to the form of the melt season initialisation demonstrates how complicated this problem is to resolve in models.

This was a comment from the other reviewer too and we agree to the point. We added some details in the manuscript to highlight this point.

The implications for large-scale projections of Greenland's behaviour (and ultimately contribution to sea level rise) could also be discussed in more detail.

We tried to give what feels like realistic implications of our results on a longer timescale in the discussion and conclusion. We however do not want to go to much farther as due to the complexity of the system it is not straightforward to extrapolate our results and long term simulations should be performed to answer those questions in more details

**2.2   Minor comments and technical corrections**

- L16-19: first couple of sentences of the introduction are a little repetitive – suggest combining into one sentence.

  This has been rephrased

- L24: "late 70's " → "late 1970s"

  Done

- L43: space needed between subglacial and drainage

  Done

- L78: probably worth briefly defining effective pressure in this context

  This has been added as part of the more detailed description of the hydrological model as asked by Reviewer #1

- L108: First place that ERA5 is mentioned and so needs more of an introduction.

  The mention to ERA5 have been removed here as it was not strictly necessary and it is described in more details in the "Forcing" section.

- L111: quantify "small volume loss"

  Done

- L118-122: I'm not sure what the Cryosphere style guide is, but I would change the bullet point symbol to something else

  I am not sure either but we dropped the symbols altogether.

- Eqs 5, 6: don't italicise max

  Fixed

- Section 2.4: On second reread it makes more sense, but what is the outcome of performing the Wilcoxon signed-rank test? Do you reject ensemble members that are significantly different from the reference? How does this relate to the analysis of the subsequent experiments?

  No we keep all ensemble members, the Wilcoxon test is only used to define significant difference with respect to the reference simulation in the table relative to each experiment. We clarified this point in the text.

- L186-7: Last sentence of this paragraph should be in the next paragraph (about local effects)

  The text has been modified accordingly

- L190-200: Could changes in geometry (specifically surface slope) also contribute to the propagation of acceleration upstream over the season? I.e. the initial acceleration at lower elevations causes a steepening of the ice surface just upstream resulting in an increase in driving stress. Perhaps the effect is very small compared to the impact of changes in N, but we see this diffusive response after retreat events in marine terminating glaciers.

  I am not actually sure of how that would impact the glacier. I expect that this would have a very small impact on velocities as the geometry change is minor.

- Fig 4: define horizontal black line in caption

  Added

- L222: bellow → below

  Corrected

- L239: "short melt season" → should this instead be the "long melt season"?

  Yes, that has been fixed

- L243: What is the EPL?

  It is an other personal acronym for EDS, it has been replaced in the text

- L244-246: It is unclear exactly what is meant by the "overshoot" – specify in which variable, and which figure. Only Fig 5e-f is mentioned here, but I think you are referring to the later summer increase in N to a value above the winter average, shown in Fig 5c only.

The reference to Fig 5e-f here was pointing towards the more developed efficient drainage system. We reformulated this sentence to make it clearer

- Fig 8: I like this figure, although looking at panel a, there appears to be a slight offset in runoff between the reference simulation and the other two simulations that are meant to have a constant runoff (as discussed in section 3.2). Is the offset real and if so why is it there?

  Yes this offset is real, it is due to the slight differences in the evolution of the ice thickness throughout the different simulations. The parameterisation for the melt was done on the initial geometry which explains this slight drift.

- Section 3.5: suggest incorporating this section into the discussion.

  This section has been reworked into the discussion.

- L379: "The results of our model [experiments/simulations] suggest..."

  We kept model here, our intent is that one might get different results with a different model as pointed in the shortcoming section.

- L397-8: "It must be noted however..." could you clarify this sentence? Perhaps "velocities averaged over the season" or "seasonally averaged velocities" rather than "seasonal velocities"

  Done

---

## Author Response (AR2)

**Answer to the editor's comments on our manuscript entitled : "Impact of runoff temporal distribution on ice dynamics."**

Basile de Fleurian, Richard Davy and Petra M. Langebroek

**1 Overall comments**

I appreciate the enhanced description of the dual-continuum model (lines 90-102). However, it is problematic because it is word-for-word the same as the description in de Fleurian et al. (2016). This portion of the manuscript needs thorough paraphrasing before it can be published in The Cryosphere.
This was in fact a copy past of the preceding description of the model and my impression was that it is fine in this type of context to reuse the description as it is still the same model. I rephrased it anyway so it should hopefully not be too similar anymore.

Figure 1 inset – It would be clearer to understand if the y-axis were "Runoff" instead of "Temperature". I get that you used a PDD so your base forcing truly is temperature, but the processes that you are studying relate to runoff, and runoff is more frequently discussed in your text (as well as shown as the forcing in all other figure panels) than temperature.
Done

Figure 4 should include altitude values on panels b and c as well, not just panel d.
Done

**2 Minor Changes**

- Abstract remove or rephrase "surprisingly" as it is imprecise / unscientific. Similarly, remove or rephrase "and/or" as it is informal.
  Done

- Clarify which of the sensitivity runs (low / medium / high runoff) is the reference simulation. It is fairly intuitive that it should be the medium one, but this is never stated.

Actually table 2 states "reference value" and not medium value. This has been emphasised also at the beginning of section 3.1 for clarity.

- The elevation of the runoff limit is unclear. I understand that it varies between runs (low, medium, and high runoff sensitivity runs). Presenting this information in either text, a table (perhaps Table 2), or on figures would be helpful, as I found myself searching the manuscript for the runoff limit. From Figure 3 I could infer that it lies between 1215 m and 1465 m for the reference simulation.
  A sentence has been added in the forcing section to give those numbers

- SSUmax should be defined specifically. Currently it just says "this shift in behaviour" (line 234), although I think that SSUmax is actually an elevation. What SSUmax stands for should be defined, and what the "max" refers to elevation or speed? should be stated as well.
  The sentence on line 234 has been rephrased to clarify that point

- line 306-7 a few extra words here, "the" and "to"
  Fixed

- line 328 "more larger" should be corrected
  Corrected

- Table 5 Instead of "Colder temperatures", it should say more directly what the experiment actually was: "lower intensity of runoff" (or something similar), and same for "Warmer temperatures".
  The experiments are actually based on warmer or colder temperature but we agree that higher or lower intensity makes more sense in this manuscript.

- line 361 similar to the comment above, replace "colder"
  Changed.

- line 379 instead of "strong" velocities, this should say "high" or "fast"
  Replaced.

- line 385 rephrase "rather inefficient efficient draining system"
  This sentence has been rephrased.

- line 403 rephrase the sentence starting "However, we think..." in order to make it more clear which you are hypothesizing (that there is a reasonable physical explanation) and which is fact rather than speculation (that your results are robust).
  This has been rephrased.

- line 454 "pauses" should be "poses"
  Changed.

- line 462 specify that you mean –spatial– distribution here.
  Fixed.
  Further along in this paragraph, you mention that moulin location is likely unimportant, but would you care to address moulin density (e.g. Banwell et al., 2016)? Essentially, your model has a moulin at every grid node, so it is maximally dense.
  Reading again this sentences it actually feels that moulin density is actually more suited here than moulin position, the sentence has been corrected accordingly.
  Finally, the last three sentences in this paragraph (lines 469-473) are a bit convoluted and should be smoothed out to make your point clear: that the highest injection point in real life is not likely to change from year to year. As currently written, the reasoning switches back and forth with "however"s that seem to me to be misleading (i.e. not actually in contradiction with the previous point). Perhaps this made me misunderstand the basic argument. Please check and revise accordingly.
  This part have been rephrased

- line 475 "without performing the actual simulations" do you mean simulations with real-life runoff inputs?
  No this was more related to the length of the simulation that would be needed to evaluate the response of the coupled system on a longer term, this has been clarified.

- line 501 change "constant runoff" to something that specifies that the time-integrated runoff volume is unchanged from simulation to simulation. As is, "constant runoff" could be read as unchanging in time.
  This has been clarified.

- line 502 change "originally" to "initially"
  Done.

- line 512 "led" seems like not the right word here and is redundant with "at the beginning of the melt season"
  I could not spot the redundancy but changed "led" to "performed" which seemed to be better suited.

- line 514 I believe you're actually advocating for use of models with –both– inefficient and efficient components, yes? As written now, it could be interpreted that a channel-only model is a good approach. Please revise.
  Yes I am indeed advocating for double components models, that as been reformulated

---

## Author Response (AR3)

**Answer to the editor's comments on our manuscript entitled : "Impact of runoff temporal distribution on ice dynamics."**

Basile de Fleurian, Richard Davy and Petra M. Langebroek

Thank you for this comment on the preceding version of the manuscript. The paragraph treating of the various potential shortcomings of our study has been rewritten in three different paragraphs as suggested. Hopefully this part is now clearer.

Thank you again for your thorough reading of this manuscript

Basile de Fleurian